# Myopalladin knockout mice develop cardiac dilation and show a maladaptive response to mechanical pressure overload

Maria Carmela Filomena[1,2], Daniel L Yamamoto[1], Pierluigi Carullo[1,2], Roman Medvedev[2,3†], Andrea Ghisleni[4‡], Nicoletta Piroddi[5], Beatrice Scellini[5], Roberta Crispino[6], Francesca D'Autilia[2], Jianlin Zhang[7], Arianna Felicetta[2,8], Simona Nemska[2], Simone Serio[2], Chiara Tesi[5], Daniele Catalucci[2], Wolfgang A Linke[9], Roman Polishchuk[6], Corrado Poggesi[5], Mathias Gautel[4], Marie-Louise Bang[1,2]*

[1]Institute of Genetic and Biomedical Research (IRGB) - National Research Council (CNR), Milan unit, Milan, Italy; [2]IRCCS Humanitas Research Hospital, Milan, Italy; [3]Department of Cardiac Surgery, University of Verona, Verona, Italy; [4]Randall Centre for Cell and Molecular Biophysics, King's College London BHF Centre of Research Excellence, London, United Kingdom; [5]Department of Experimental and Clinical Medicine, University of Florence, Florence, Italy; [6]Telethon Institute of Genetics and Medicine (TIGEM), Pozzuoli, Italy; [7]Department of Medicine, University of California, San Diego, La Jolla, United States; [8]Humanitas University, Pieve Emanuele, Italy; [9]Institute of Physiology II, University of Muenster, Muenster, Germany

*For correspondence:
marie-louise.bang@cnr.it

Present address: †Department of Medicine, University of Wisconsin, Madison, United States; ‡FIRC Institute of Molecular Oncology (IFOM), Milan, Italy

**Abstract** Myopalladin (MYPN) is a striated muscle-specific immunoglobulin domain-containing protein located in the sarcomeric Z-line and I-band. *MYPN* gene mutations are causative for dilated (DCM), hypertrophic, and restrictive cardiomyopathy. In a yeast two-hybrid screening, MYPN was found to bind to titin in the Z-line, which was confirmed by microscale thermophoresis. Cardiac analyses of MYPN knockout (MKO) mice showed the development of mild cardiac dilation and systolic dysfunction, associated with decreased myofibrillar isometric tension generation and increased resting tension at longer sarcomere lengths. MKO mice exhibited a normal hypertrophic response to transaortic constriction (TAC), but rapidly developed severe cardiac dilation and systolic dysfunction, associated with fibrosis, increased fetal gene expression, higher intercalated disc fold amplitude, decreased calsequestrin-2 protein levels, and increased desmoplakin and SORBS2 protein levels. Cardiomyocyte analyses showed delayed $Ca^{2+}$ release and reuptake in unstressed MKO mice as well as reduced $Ca^{2+}$ spark amplitude post-TAC, suggesting that altered $Ca^{2+}$ handling may contribute to the development of DCM in MKO mice.

## Introduction

The Z-line of striated muscle is a highly organized multiprotein complex at the boundary between sarcomeres, connecting the contractile apparatus with the cytoskeleton and extracellular matrix. Within the Z-line, actin and titin filaments from adjacent sarcomeres are cross-linked by α-actinin, which also binds to a number of other Z-line proteins (*Bang, 2017*; *Frank and Frey, 2011*; *Knöll et al., 2011*). Thus, the Z-line is important for efficient force production and transmission of force both between sarcomeres and laterally from the Z-line to adjacent myofibrils, the sarcolemma, and the extracellular matrix. In addition to providing structural stability and maintaining the structural integrity of muscle during contraction, the Z-line plays an essential role in mechanotransduction by translating

biomechanical stress into biochemical signals, important for the maintenance of muscle homeostasis and adaptation to altered mechanical load. In response to biomechanical stress, such as pressure or volume overload, the heart undergoes compensatory cardiac hypertrophy, allowing the heart to maintain its function in conditions of increased workload (*Diwan and Dorn, 2007*; *Kehat and Molkentin, 2010*). However, sustained stress ultimately leads to maladaptive remodeling and transition to heart failure. Z-line proteins play important roles in the adaptation to increased biomechanical stress and mutations in many Z-line-associated proteins have been linked to cardiomyopathies (*Bang, 2017*; *Frank and Frey, 2011*; *Knöll et al., 2011*).

Myopalladin (MYPN) is a 145 kDa striated muscle-specific sarcomeric protein belonging to a small family of actin-associated immunoglobulin (Ig) domain-containing proteins in the Z-line, comprising MYPN, palladin (PALLD), and myotilin (*Otey et al., 2009*; *Otey et al., 2005*). MYPN contains five Ig domains and is present in both the nucleus and the sarcomere, where it has a dual localization in the Z-line and the I-band. MYPN binds to various Z-line proteins in the heart, including α-actinin (*Bang et al., 2001*), nebulette (*Bang et al., 2001*), and the PDZ-LIM family members Cypher/ZASP, CLP36, ALP, and RIL (*von Nandelstadh et al., 2009*). In addition, it interacts with the transcriptional cofactor cardiac ankyrin repeat protein (CARP/Ankrd1), which shuttles between the nucleus and the I-band, where it is linked to the titin N2A region, potentially implicating it in mechano-dependent signaling (*Bang et al., 2001*; *Miller et al., 2003*). CARP is strongly induced under various conditions of stress (reviewed in *Bang et al., 2014*; *Ling et al., 2017*) and has been reported to negatively regulate the expression of cardiac genes, such as *Nppa*, *Myl2*, and *Tnnc1* (*Jeyaseelan et al., 1997*; *Kuo et al., 1999*; *Zou et al., 1997*). Finally, we recently demonstrated that MYPN, like its other family members, binds to filamentous actin (F-actin), preventing actin depolymerization (*Filomena et al., 2020*). Furthermore, it binds to myocardin-related transcription factor A and B (MRTF-A and MRTF-B), which shuttle between the cytosol and the nucleus in response to alterations in actin dynamics and act as cofactors for serum response factor (SRF), controlling its activity (*Filomena et al., 2020*).

The essential role of MYPN for normal cardiac function is supported by the identification of an increasing number of heterozygous mutations in the *MYPN* gene associated with various types of human cardiomyopathy, including hypertrophic (HCM), dilated (DCM), and restrictive (RCM) cardiomyopathy (*Bagnall et al., 2010*; *Duboscq-Bidot et al., 2008*; *Meyer et al., 2013*; *Purevjav et al., 2012*). The human phenotype could be partially recapitulated in animal models, including transgenic mice for the p.Y20C variant (*Purevjav et al., 2012*), which has been correlated with DCM and HCM, as well as heterozygous knockin mice for the p.Q529X nonsense mutation associated with RCM (*Huby et al., 2014*; *Purevjav et al., 2012*). However, the underlying disease mechanisms are poorly understood and the functional role of MYPN in the heart has remained elusive. To provide further insights into the role of MYPN in the heart, we studied the cardiac phenotype of MYPN knockout (MKO) mice in which we recently characterized the skeletal muscle phenotype (*Filomena et al., 2020*). In particular, we found that MKO mice develop mild cardiac dilation and progressive systolic dysfunction as well as show a maladaptive response to mechanical pressure overload by transaortic constriction (TAC), developing severe cardiac dilation and systolic dysfunction, associated with increased fetal gene expression, more convoluted intercalated discs (ICDs), and altered $Ca^{2+}$ handling. Furthermore, in biochemical studies, we identified a novel direct interaction between MYPN and titin in the Z-line.

## Results

### The MYPN C-terminal region binds to the titin Ig domains Z4-Z5 in the Z-line

In a yeast two-hybrid (Y2H) screening of a skeletal muscle cDNA library using titin Ig domains Z4-Z5 in the titin Z-line region as a bait, we identified MYPN as an interaction partner (*Figure 1A*) with the longest clones starting at amino acid 814 (Acc. NM_032578.3). We subsequently narrowed down the interaction site to the MYPN C-terminal region from residue 935 to 1320, containing its three C-terminal Ig domains. The corresponding region in its homologue PALLD (Res. 794–1123; Acc. NM_001166108) likewise bound to titin. To confirm the interactions, we performed a microscale thermophoresis binding (MST) assay (*Seidel et al., 2013*). Initially, NT-647-labeled titin Ig domain Z4-Z5 peptide was titrated with increasing concentrations of the MYPN C-terminal fragment as well as the homologous fragment of PALLD (*Figure 1B*). However, due to protein aggregation at higher

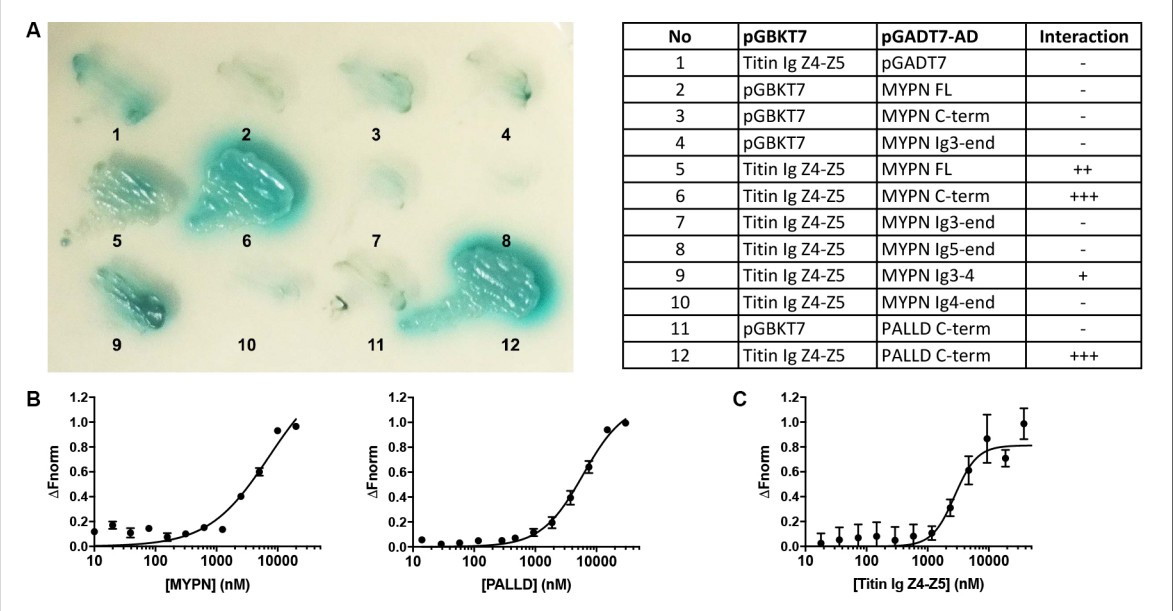

**Figure 1.** Myopalladin (MYPN) directly binds to titin in the Z-line. (**A**) Yeast two-hybrid (Y2H) assay showing binding of the C-terminal region of MYPN and its homologue palladin (PALLD) to the titin Ig domains Z4-Z5 (titin Ig Z4-Z5) in the Z-line. A culture plate with different combinations of Y2H co-transformations is shown as indicated in the table. (**B**) Microscale thermophoresis (MST) analysis of 450 nM labeled titin Ig Z4-Z5 incubated with increasing concentrations of MYPN and PALLD C-terminal domains. Due to precipitation, signal saturation was not reached. The dose/response curve is representative of two independent experiments (n = 17 and n = 16 runs). Fnorm, normalized fluorescence. (**C**) MST analysis of 400 nM labeled MYPN C-terminal domain incubated with increasing concentrations of titin Ig Z4-Z5. The dose/response curve is representative of four independent experiments (n = 7 runs).

The online version of this article includes the following source data for figure 1:

**Source data 1.** Microscale thermophoresis binding (MST) analysis of myopalladin (MYPN)/titin interaction.

concentrations, it was not possible to exceed MYPN and PALLD concentrations of 20 and 30 µM, respectively, at which full binding saturation had not yet been completely reached. For that reason, the equilibrium dissociation constants ($K_d$) could not be unequivocally determined, but were estimated to 7.2 ± 2.6 µM for MYPN and 6.1 ± 0.8 µM for PALLD. This experimental limitation was circumvented by generation of NT-647-labeled MYPN at dilutions in the lower nanomolar range and titration with increasing concentrations of the more stable titin Ig domain Z4-Z5 fragment. Under this experimental condition, full binding saturation was achieved and the $K_d$ was estimated to 2.8 ± 0.5 µM (**Figure 1C**). This revealed that the C-terminal region of MYPN as well as the homologous PALLD region bind to the titin Ig domains Z4-Z5 with apparent affinities in the low micromolar range, comparable to other Z-line scaffold complexes (e.g. the interaction between titin Z-repeats and ACTN2), which have $K_d$ values between 100 nM and 4 µM (**Joseph et al., 2001**).

## Histological and ultrastructural analyses of MKO mouse hearts

To determine the structural and functional role of MYPN in the heart in vivo, we took advantage of constitutive MKO mice, generated by gene targeting as recently described (**Filomena et al., 2020**). As previously reported, MKO mice are born at Mendelian ratios and have a normal life span (**Filomena et al., 2020**). Histological analyses by hematoxylin and eosin (H&E) staining of hearts from wild-type (WT) and MKO mice revealed no histological abnormalities up to 6 months of age and Picro Sirius Red stainings showed no evidence of fibrosis, necrosis, or myofibrillar disarray in WT and MKO mice (**Figure 2A**). Furthermore, TUNEL staining showed no signs of apoptosis (data not shown). Measurements on isolated cardiomyocytes (CMCs) from 4-month-old mice showed similar CMC length in WT and MKO mice (**Figure 2B and C**), but significantly reduced CMC width and size in MKO mice (**Figure 2D and E** and **Figure 2—figure supplement 1**), possibly due to reduced body weight of MKO mice (11.8 % at 4 months of age; see **Figure 3J** and **Figure 3—source data 2**) as previously reported (**Filomena et al., 2020**). The body weight to tibia length ratio was reduced in MKO mice (**Figure 2F**),

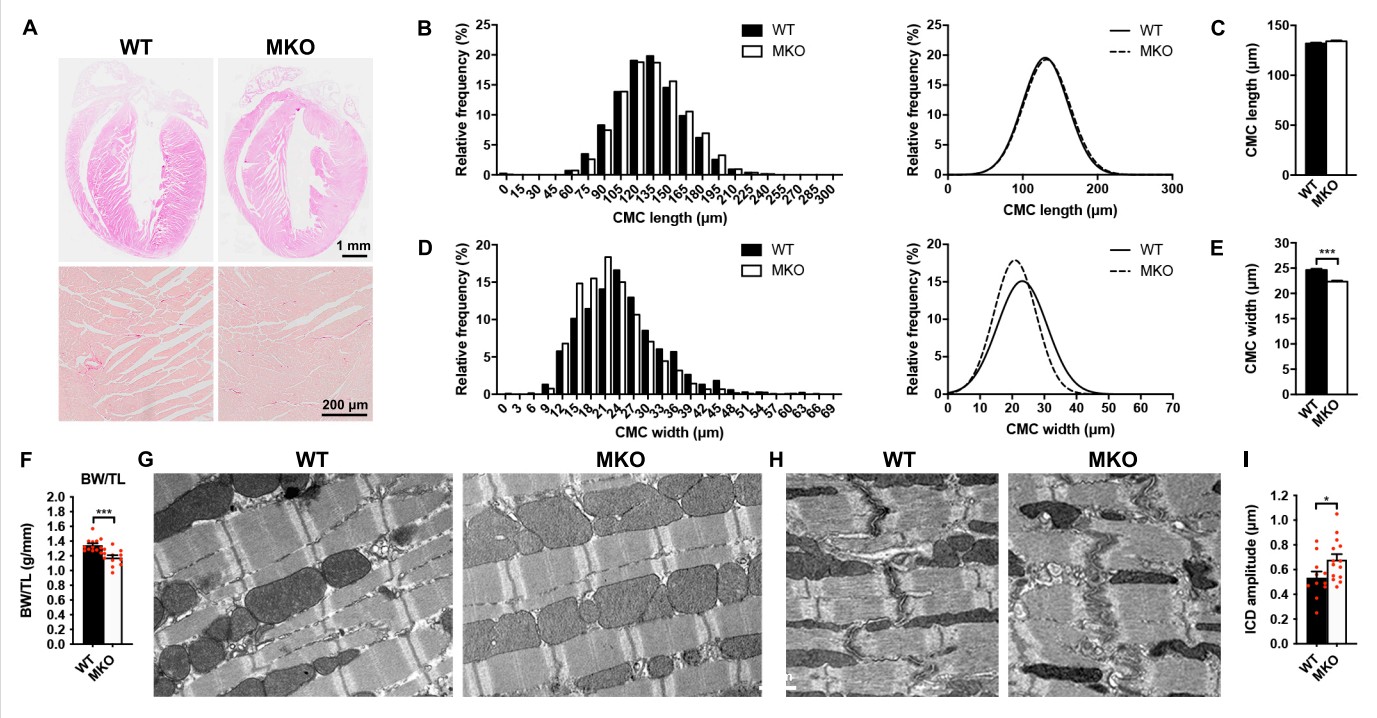

**Figure 2.** Histological and electron microscopy analyses of hearts from wild-type (WT) and myopalladin knockout (MKO) mice. (**A**) Hematoxylin and eosin (H&E) (top) and Picro Sirius Red (bottom) stainings showing no morphological abnormalities or fibrosis in the heart of 6-month-old MKO male mice compared to WT mice (n = 3 per group). (**B**) Histograms and Gaussian distribution curves showing similar adult cardiomyocyte (CMC) length in 4-month-old WT and MKO male mice (n = 1292 CMCs from three WT mice and 1740 CMCs from three MKO mice). (**C**) Average CMC length in WT and MKO male mice. Data are represented as mean ± standard error of the mean (SEM). (**D**) Histograms and Gaussian distribution curves showing reduced adult CMC width in 4-month-old MKO male mice compared to WT mice (n = 1373 CMCs from three WT mice and 1193 CMCs from three MKO mice). (**E**) Average CMC width in WT and MKO male mice. Data are represented as mean ± SEM. ***$p < 0.001$; unpaired Student's t-test. (**F**) Body weight to tibia length ratio (BW/TL) of 10-week-old WT and MKO male mice. Data are represented as mean ± SEM (n = 12–13 per group). ***$p < 0.001$; unpaired Student's t-test. (**G–H**) Electron micrographs of papillary muscle from 8-month-old WT and MKO male mice, showing sarcomere (**G**) and intercalated disc (ICD) (**H**) structure. (**I**) Average ICD fold amplitude in WT and MKO mice. Data are represented as mean ± SEM (n = 11 ICDs from WT mice and 14 ICDs from MKO mice). *$p < 0.05$; unpaired Student's t-test.

The online version of this article includes the following source data and figure supplement(s) for figure 2:

**Source data 1.** Cardiomyocyte (CMC) size measurements in wild-type (WT) and myopalladin knockout (MKO) male mice.

**Source data 2.** Body weight to tibia length ratio (BW/TL) measurements in wild-type (WT) and myopalladin knockout (MKO) male mice.

**Source data 3.** Intercalated disc (ICD) fold amplitude measurements in wild-type (WT) and myopalladin knockout (MKO) male mice.

**Figure supplement 1.** Cardiomyocyte (CMC) size in 4-month-old wild-type (WT) and myopalladin knockout (MKO) male mice.

indicating that the lower body weight of MKO mice is mainly due to reduced muscle weight as a result of the previously reported reduction in myofiber cross-sectional area (*Filomena et al., 2020*). Transmission electron microscopy analyses of papillary muscle from 8-month-old MKO compared to littermate control mice showed normal sarcomere organization with no alterations in Z-line structure (*Figure 2G*). Also ICD structure appeared normal, but measurement of ICD fold amplitude revealed an ~ 27 % increase in MKO mice compared to WT mice (*Figure 2H1*).

## Echocardiographic analysis reveals progressive cardiac dilation and systolic dysfunction in MKO mice

The effect of MYPN ablation on cardiac morphology and function was evaluated by echocardiography of WT and MKO male mice at 10 weeks, 4 months, and 6 months of age (*Figure 3A–I* and *Figure 3—source data 1*). This revealed mild left ventricular dilation without alteration in left ventricular wall and septum thickness in MKO mice, consistent with eccentric remodeling. This was associated with a

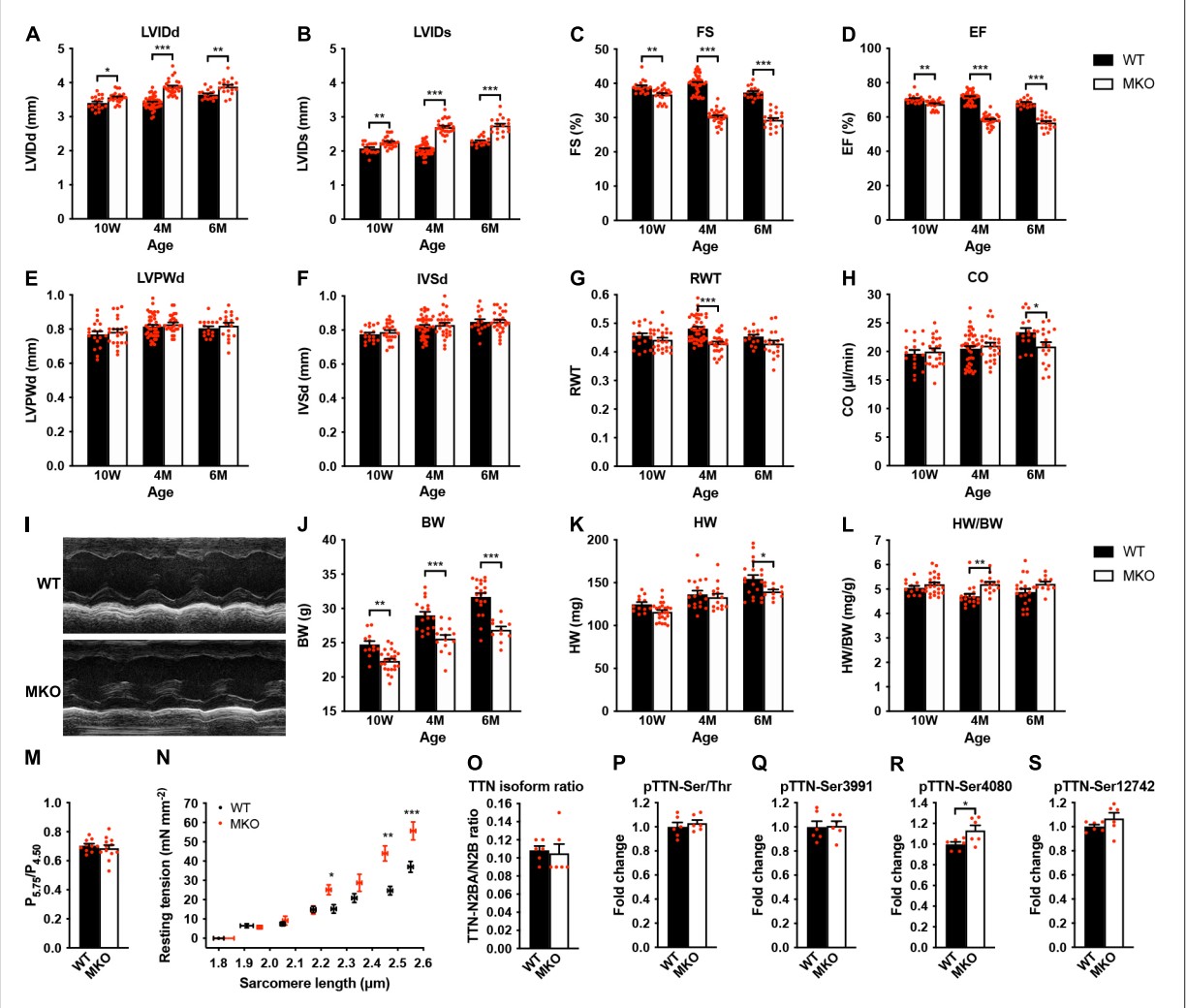

**Figure 3.** Echocardiographic analyses and sarcomere-length-tension relationship in cardiac myofibrils from wild-type (WT) and myopalladin knockout (MKO) mice. (**A–H**) Echocardiographic analysis of WT and MKO male mice at 10 weeks (10 W), 4 months (4 M), and 6 months (6 M) of age. LVID, left ventricular inner diameter; FS, fractional shortening; EF, ejection fraction; LVPW, left ventricular posterior wall thickness; IVS, interventricular septum thickness; RWT, relative wall thickness ((LVPWd + IVSd)/LVIDd); CO, cardiac output; d, diastole; s, systole. Data are represented as mean ± standard error of the mean (SEM) (n = 16–42 per group). *p < 0.05; **p < 0.01; ***p < 0.001; two-way analysis of variance (ANOVA) with Bonferroni's multiple comparison test. (**I**) Representative echocardiographic short-axis M-mode images from hearts of 6-month-old WT and MKO male mice. (**J–L**) Body weight (BW) (**J**), heart weight (HW) (**K**), and heart weight to body weight ratio (HW/BW) (**L**) of WT and MKO male mice at 10 W, 10 M, and 6 M of age. Data are represented as mean ± SEM (n = 12–24 per group). *p < 0.05; **p < 0.01; ***p < 0.001; two-way ANOVA with Bonferroni's multiple comparison test. (**M**) Ratio of tension measured at pCa 5.75 ($P_{5.75}$, submaximal activation) vs. pCa 4.50 ($P_{4.50}$, maximal activation) in WT and MKO myofibrils using $Ca^{2+}$ jump protocols. Data are represented as mean ± SEM (n = 10 myofibrils from two WT mice and 12 myofibrils from two MKO mice). (**N**) Average sarcomere-length-tension relationship in cardiac myofibrils from the left ventricle of 4-month-old WT and MKO male mice. Each data point is represented as mean ± SEM from 10 myofibrils from three WT mice and 16 myofibrils from three MKO mice. *p < 0.05; **p < 0.01; ***p < 0.001; unpaired Student's t-test. (**O–S**) Densitometric analysis for (**O**) titin (TTN) N2BA/N2B isoform ratio as determined by sodium dodecyl sulfate (SDS)-PAGE and Coomassie blue staining, (**P**) titin serine/threonine phosphorylation as determined by Western blot analysis using anti-phosphoserine/threonine antibody, and (**Q–R**) site-specific titin phosphorylation on Ser3991 (corresponding to human pTTN-Ser4010, phosphorylated by PKA and ERK2) (**Q**), Ser4080 (corresponding to human pTTN-Ser4099, phosphorylated by PKG) (**R**), and Ser12742 (corresponding to human pTTN-Ser11878, phosphorylated by PKCα) (**S**) using titin phosphosite-specific antibodies. Normalization was performed to total protein content as determined by Coomassie blue staining of each blot. Data are represented as mean ± SEM (n = 6 per group). *p < 0.05; unpaired Student's t-test.

The online version of this article includes the following figure supplement(s) for figure 3:

**Figure supplement 1.** Titin isoform expression and phosphorylation in the left ventricle of 4-month-old wild-type (WT) and myopalladin knockout (MKO) male mice.

**Source data 1.** Echocardiographic parameters of wild-type (WT) and myopalladin knockout (MKO) male mice at different ages.

*Figure 3 continued on next page*

Figure 3 continued

**Source data 2.** Echocardiographic analyses and heart weight to body weight ratio (HW/BW) measurements on wild-type (WT) and myopalladin knockout (MKO) male mice under basal conditions.

**Source data 3.** Heart weight to body weight (HW/BW) measurements on wild-type (WT) and myopalladin knockout (MKO) male mice under basal conditions.

**Source data 4.** Sarcomere-length-tension relationship and calcium jump experiments in cardiac myofibrils from wild-type (WT) and myopalladin knockout (MKO) male mice.

**Source data 5.** Densitometry of titin blots.

reduction in cardiac systolic function in MKO mice as reflected by a fractional shortening of 30.2% ± 0.4 % in MKO mice vs. 40.0% ± 0.4 % in WT mice at 4 months of age as well as a significant reduction in cardiac output at 6 months of age (20.9 ± 0.9 in MKO mice vs. 23.4 ± 0.7 in WT mice). Heart weight was reduced in MKO mice at 6 months of age, but due to reduced body weight of MKO mice, there was a trend toward increased heart weight to body weight ratio in MKO mice, which, however, was only statistically significant in 4-month-old mice (*Figure 3J–L* and *Figure 3—source data 1*).

## Decreased isometric tension and increased resting tension at longer sarcomere lengths in MKO myofibrils

To determine the effect of MKO on sarcomere mechanics, we compared biomechanical properties of myofibril preparations from the left ventricle of 4-month-old WT and MKO mice. Consistent with the reduced cardiac systolic function observed in MKO mice, maximum $Ca^{2+}$-activated isometric tension was reduced in MKO myofibrils, while there were no differences in the kinetics of force generation and relaxation following rapid $Ca^{2+}$ increase and removal (*Table 1*). The lack of effect of MYPN ablation on force kinetics suggests that the impairment of active force generation in MKO mice is not due to changes in the number of force generating cross-bridges. Furthermore, measurements of the tension generated at submaximal (pCa 5.75) vs. maximal (pCa 4.50) $Ca^{2+}$ activation in $Ca^{2+}$ jump experiments showed a similar ratio in WT and MKO myofibrils (*Figure 3M*), providing strong evidence that the lack of MYPN does not affect myofilament $Ca^{2+}$ sensitivity. The resting tension at optimal sarcomere length also did not differ between WT and MKO preparations (*Table 1*). On the other hand, determination of the sarcomere-length-tension relationship revealed increased resting tension in MKO myofibrils at sarcomere lengths > 2.20 µm (*Figure 3N*).

The passive mechanical properties of cardiac myofibrils are mostly determined by titin (*Linke and Fernandez, 2002*). Titin-based stiffness is regulated by titin isoform expression (i.e. the ratio between the long and compliant titin N2BA isoform and the shorter and stiffer titin N2B isoform) and post-translational modifications of the elastic I-band titin region, including phosphorylation and oxidative modifications, such as *S*-glutathionylation and intramolecular disulfide bond formation and isomerization (*Koser et al., 2019*). In particular, hypophosphorylation of the N2B unique sequence (N2Bus) and hyperphosphorylation of the PEVK domain increase passive stiffness. Therefore, to determine whether MKO myofibrils may exhibit increased titin-based stiffness, we determined titin isoform expression (titin N2BA/N2B ratio) (*Figure 3O* and *Figure 3—figure supplement 1*) as well as both

**Table 1.** Tension generation and relaxation in ventricular myofibrils from wild-type (WT) and myopalladin knockout (MKO) male mice.

| | Resting conditions | | Tension generation | | Relaxation | | |
| | | | | | Slow phase | | Fast phase |
| Myofibril batch | SL (µm) | RT (mN/mm²) | $P_0$ (mN/mm²) | $k_{ACT}$ (s⁻¹) | Duration (ms) | $k_{REL}$ (s⁻¹) | $k_{REL}$ (s⁻¹) |
|---|---|---|---|---|---|---|---|
| WT | 2.22 ± 0.02 (26) | 11.1 ± 1.4 (26) | 161 ± 11 (26) | 5.17 ± 0.29 (23) | 74 ± 4 (20) | 1.96 ± 0.18 (20) | 47 ± 4 (23) |
| MKO | 2.21 ± 0.01 (27) | 10.3 ± 1.3 (26) | 112 ± 6***(27) | 5.38 ± 0.40 (26) | 72 ± 3 (22) | 2.03 ± 0.15 (19) | 30 ± 4 (24) |

All values are presented as mean ± standard error of the mean (SEM). Numbers in parentheses are number of myofibrils. SL, sarcomere length, RT, resting tension, $P_0$, maximum isometric tension; $k_{ACT}$, rate constant of tension rise following step-wise pCa decrease (8.0→4.5) by fast solution switching. Full tension relaxation kinetics were characterized by the duration and rate constant of tension decay of the isometric slow relaxation phase (slow $k_{REL}$) and the rate constant of the fast relaxation phase (fast $k_{REL}$). ***p < 0.001 vs. WT; unpaired Student's t-test.

total and site-specific phosphorylation by Western blot analyses using a phosphoserine/threonine-specific antibody (*Figure 3P* and *Figure 3—figure supplement 1*) and phospho-specific antibodies against Ser3991 (phosphorylated by PKA and ERK2) and Ser4080 (phosphorylated by PKG) in the N2Bus region and Ser11742 (phosphorylated by PKCα) in the PEVK region, respectively (*Figure 3* and *Figure 3—figure supplement 1*). However, no differences were observed between WT and MKO mice except for a slight increase in Ser4080 phosphorylation (which reduces passive stiffness) in MKO mice (*Figure 3R* and *Figure 3—figure supplement 1*), suggesting that the increased stiffness in MKO myofibrils is not due to increased titin-based stiffness, although it cannot be excluded that titin may be differentially phosphorylated on other sites within its elastic spring elements or subject to oxidative modification.

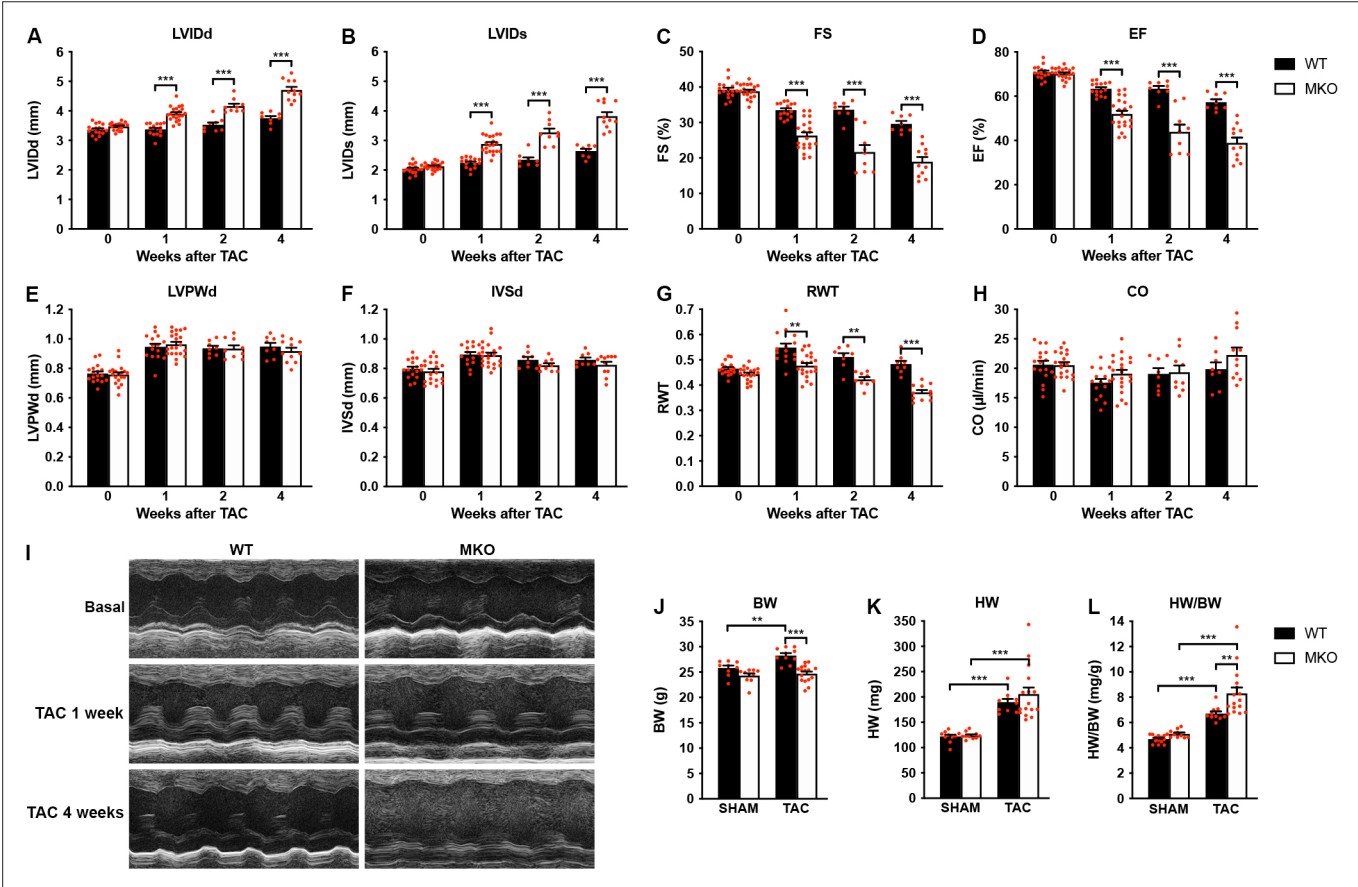

**Figure 4.** Echocardiographic analyses of wild-type (WT) and myopalladin knockout (MKO) mice following transaortic constriction (TAC). (**A–H**) Echocardiography analyses of WT and MKO male mice under basal conditions and 1, 2, and 4 weeks after TAC. Pressure gradient >70 mmHg. LVID, left ventricular inner diameter; FS, fractional shortening; EF, ejection fraction; LVPW, left ventricular posterior wall thickness; IVS, interventricular septum thickness; RWT, relative wall thickness ((LVPWd + IVSd)/LVIDd); CO, cardiac output; d, diastole; s, systole. Data are represented as mean ± standard error of the mean (SEM) (n = 8–19 per group). **p < 0.01; ***p < 0.001; linear mixed model (LMM) with Bonferroni's multiple comparison test. (**I**) Representative echocardiographic short-axis M-mode images from hearts of WT and MKO male mice under basal conditions and 1 and 4 weeks after TAC. (**J–L**) Body weight (BW) (**J**), heart weight (HW) (**K**), and heart weight to body weight ratio (HW/BW) (**L**) of WT and MKO male mice 4 weeks after TAC or SHAM surgery. Data are represented as mean ± SEM (n = 9–11 per group). **p < 0.01; ***p < 0.001; two-way analysis of variance (ANOVA) with Bonferroni's multiple comparison test.

The online version of this article includes the following figure supplement(s) for figure 4:

**Source data 1.** Echocardiographic parameters of 8-week-old wild-type (WT) and myopalladin knockout (MKO) male mice before and after transaortic constriction (TAC).

**Source data 2.** Echocardiographic analysis on wild-type (WT) and myopalladin knockout (MKO) male mice subjected to transaortic constriction (TAC) or SHAM.

**Source data 3.** Measurements of heart weight to body weight ratio (HW/BW) in wild-type (WT) and myopalladin knockout (MKO) male mice subjected to transaortic constriction (TAC) or SHAM.

## MKO mouse hearts develop chamber dilation and significantly reduced cardiac function following TAC

To determine the response to biomechanical stress, 10-week-old WT and MKO mice were subjected to mechanical pressure overload by TAC. Pressure gradients were measured by Doppler echocardiography and cardiac morphology and function were evaluated by echocardiography 1, 2, and 4 weeks after the procedure (*Figure 4A–I* and *Figure 4—source data 1*). Both WT and MKO mice showed increased left ventricular wall and septum thickness after TAC. However, while, as expected, WT mice developed compensatory concentric hypertrophy, MKO mice developed eccentric hypertrophy as indicated by a significantly reduced relative wall thickness, characterized by left ventricular dilation as well as reduced fractional shortening and ejection fraction starting already from 1 week after TAC. Cardiac output was not affected, likely due to compensatory left ventricular dilation in MKO mice. As expected, the heart weight to body weight ratio (*Figure 4J–L* and *Figure 4—source data 1*) was increased in both WT and MKO mice 4 weeks after TAC. However, due to reduced body weight of MKO mice, the heart weight to body weight ratio was significantly higher in MKO mice compared to WT mice, although the heart weight was similar in WT and MKO mice. Histological analyses performed 4 weeks after TAC showed severe left ventricular dilation of MKO mice, often associated with left atrial enlargement (*Figure 5A*). Additionally, increased interstitial and perivascular fibrosis was observed in MKO mice (*Figure 5A*, *middle* and *Figure 5B*). Measurements of CMC size on wheat germ agglutinin (WGA)-stained sections showed increased CMC size in both WT and MKO mice 4 weeks after TAC (*Figure 5A, C, D*). However, the CMC size was significantly smaller in MKO mice compared to WT mice both in SHAM- and TAC-operated mice, consistent with the results from the analysis of isolated CMCs from 4-month-old mice (*Figure 2B–E* and *Figure 2—figure supplement 1*). Transmission electron microscopy analyses of papillary muscle showed normal sarcomere structure in MKO mice after TAC (*Figure 5E*). However, MKO mice showed an ~70% increase in ICD fold amplitude compared to WT mice, while no significant increase in ICD fold amplitude was observed in WT mice after TAC (*Figure 5F and G*). To summarize, MKO mice exhibit progressive moderate cardiac dilation and reduced cardiac systolic function under basal conditions, and quickly develop eccentric hypertrophy with severe systolic dysfunction following mechanical pressure overload.

## Altered calsequestrin-2, desmoplakin, and SORBS2 levels in MKO mice following TAC

To determine the pathways responsible for the maladaptive remodeling in MKO mice in response to biomechanical stress, we performed quantitative real-time PCR (qRT-PCR) and Western blot analyses before and after TAC at an early (4-day) time point before the initiation of adverse pathological remodeling and a late time point (1 month) at which MKO mice presented with overt heart failure. As expected, qRT-PCR analyses showed reactivation of fetal genes (*Nppa*, *Nppb*, *Ankrd1*, *Acta1*, and *Myh7*) in the left ventricle of both WT and MKO mice subjected to TAC (*Figure 6A–J*). However, significantly higher levels of *Ankrd1* and *Nppb* were found in MKO mice compared to WT mice already 4 days after TAC, and *Ankrd1* and *Nppa* were significantly upregulated in MKO mice 1 month after TAC. Additionally, *Myh6* was upregulated in MKO mice 4 days after TAC. The transcript level of the *Mypn* homologue *Palld* was similar in WT and MKO mice.

   To determine possible alterations in the activity of cardiac signaling pathways and protein levels of MYPN-interacting proteins and ICD proteins (*Figure 6K–P*), Western blot analyses were performed on left ventricular tissue from WT and MKO mice. As expected, CARP (encoded by *Ankrd1*) was highly upregulated after TAC, but despite increased transcript levels in MKO mice (*Figure 6E*), there was no significant difference between WT and MKO mice (*Figure 6K and M*). Furthermore, there were no changes in the activation of Akt and mitogen-activated protein kinase signaling pathways (*Figure 6K*). Western blot analyses for proteins involved in calcium handling revealed increased calsequestrin-2 (CASQ2) levels 4 weeks after TAC in both WT and MKO mice, but significantly reduced CASQ2 levels in MKO mice compared to WT mice 4 days after TAC (*Figure 6K and N*). At the transcriptional level, *Casq2* was not significantly changed in WT and MKO mice after TAC (*Figure 6J*), suggesting that CASQ2 is regulated at the post-transcriptional level. The expression of other proteins involved in calcium handling was similar between WT and MKO mice (*Figure 6K*). No significant differences were observed in levels (*Figure 6K*) or localization (*Figure 6—figure supplement 1*) of the MYPN homologue PALLD or MYPN-interacting proteins. On the other hand, Western blot analysis for components

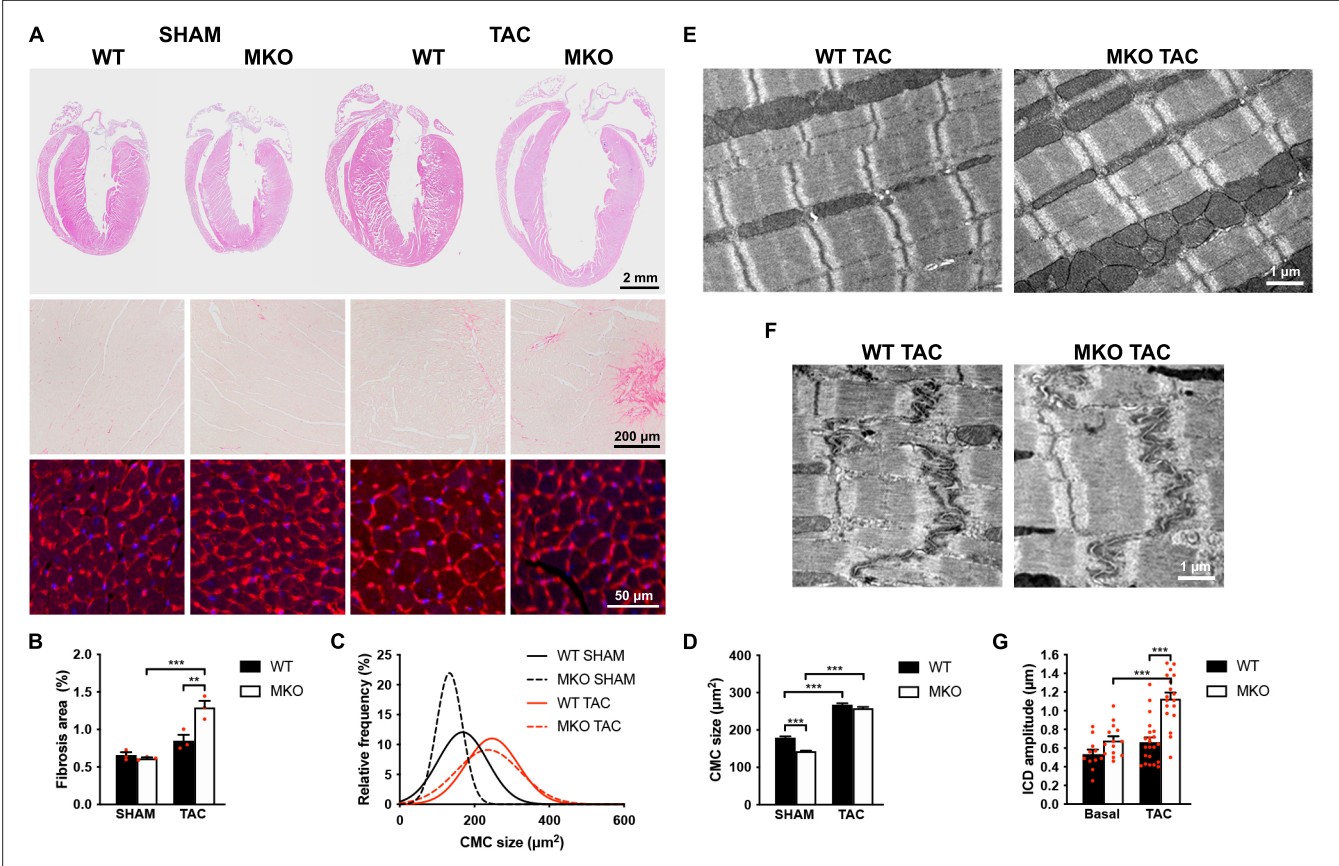

**Figure 5.** Histological and transmission electron microscopy analyses of hearts from wild-type (WT) and myopalladin knockout (MKO) male mice 4 weeks after transaortic constriction (TAC) or SHAM. (**A**) Top, Hematoxylin and eosin (H&E) stainings of hearts from WT and MKO mice subjected to TAC or SHAM. Middle, Representative Picro Sirius Red stainings of the left ventricle, showing fibrosis in MKO mice after TAC. Bottom, Representative wheat germ agglutinin (WGA) stainings of the left ventricle (red). Nuclei are visualized by DAPI (blue). (**B**) Percent area of interstitial fibrosis in the left ventricle. Data are represented as mean ± standard error of the mean (SEM) (n = 3 per group). **p < 0.01; ***p < 0.001; two-way analysis of variance (ANOVA) with Bonferroni's multiple comparison test. (**C**) Gaussian distribution curves from histograms of cardiomyocyte (CMC) size in the left ventricle of WT and MKO mice subjected to TAC or SHAM (n = 333 CMCs from three WT mice and 809 CMCs from three MKO mice subjected to SHAM; 423 CMCs from three WT mice and 665 CMCs from three MKO mice subjected to TAC). (**D**) Average CMC size in WT and MKO mice subjected to TAC or SHAM. Data are represented as mean ± SEM. **p < 0.01; ***p < 0.001; two-way ANOVA with Bonferroni's multiple comparison test. (**E–F**) Electron micrographs of papillary muscle from WT and MKO mice 4 weeks after TAC, showing sarcomere (**E**) and intercalated disc (ICD) (**F**) structure. (**G**) Average ICD fold amplitude in WT and MKO mice. Data are represented as mean ± SEM (n = 11 ICDs from WT mice and n = 14 ICDs from MKO mice, n = 22 ICDs from WT mice, and n = 18 ICDs from MKO mice subjected to TAC). ***p < 0.001; two-way ANOVA with Bonferroni's multiple comparison test.

The online version of this article includes the following figure supplement(s) for figure 5:

**Source data 1.** Measurements of fibrotic area in the left ventricle of wild-type (WT) and myopalladin knockout (MKO) male mice 4 weeks after transaortic constriction (TAC) or SHAM.

**Source data 2.** Cardiomyocyte (CMC) size measurements in wild-type (WT) and myopalladin knockout (MKO) male mice 4 weeks after transaortic constriction (TAC) or SHAM.

**Source data 3.** Intercalated disc (ICD) fold amplitude measurements in wild-type (WT) and myopalladin knockout (MKO) male mice 4 weeks after transaortic constriction (TAC).

of the ICD, including desmosomal (desmoplakin, plakoglobin/γ-catenin), adherens junction (N-cadherin, α-E-catenin, β-catenin, plakoglobin/γ-catenin, vinculin, sorbin and SH3 domain-containing 2 (SORBS2, also known as Arg-binding protein 2)), and gap junction (connexin 43) proteins, showed upregulation of desmoplakin in MKO vs. WT mice 4 days after TAC (~2.3-fold; *Figure 6K and O*), while SORBS2, a known interaction partner of PALLD (*Rönty et al., 2005*), was upregulated in MKO mice both under basal conditions (~1.6-fold) and at early and late time points after TAC (~1.8-fold 4 days after TAC and ~1.6-fold 4 weeks after TAC) (*Figure 6K and P*). The SORBS2 interaction site in

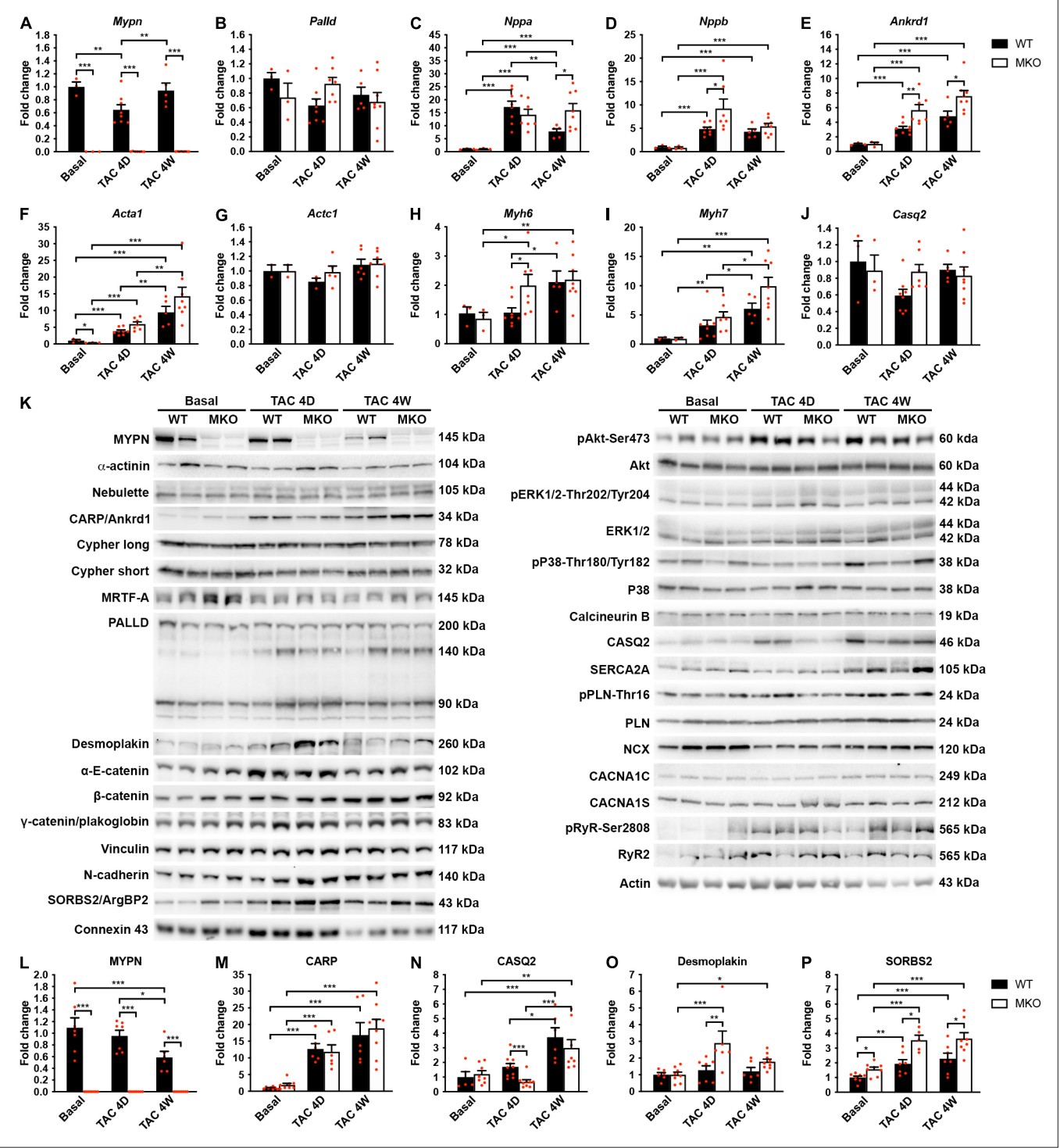

**Figure 6.** Quantitative real-time PCR (qRT-PCR) and Western blot analyses on wild-type (WT) and myopalladin knockout (MKO) mice at basal conditions and following transaortic constriction (TAC). (**A**) *qRT-PCR analysis for Mypn, Palld, and markers of cardiac remodeling on left ventricular RNA from WT and MKO male mice at basal conditions and 4 days (**D**) and 4 weeks (**W**) after TAC. B2m was used for normalization. Data are represented as mean ± standard error of the mean (SEM) (n = 3–8 per group performed in triplicate). \*p < 0.05, \*\*p < 0.01, \*\*p < 0.001; two-way analysis of variance (ANOVA) with Bonferroni's multiple comparison test. (**B**) Western blot analysis on left ventricular lysate from WT and MKO male mice at basal conditions and 4 days (**D**) and 4 weeks (**W**) after TAC. Representative blots are shown (n = 3–4 per group). (**C**) Densitometric analysis for proteins that were significantly altered in MKO mice. Normalization was performed to total protein content as assessed on TGX Stain-Free gels (Bio-Rad Laboratories). Data are represented as mean ± SEM (n = 3–4 per group). \*p < 0.05, \*\*p < 0.01, \*\*\*p < 0.001; two-way ANOVA with Bonferroni's multiple comparison test.

*Figure 6 continued on next page*

*Figure 6 continued*

The online version of this article includes the following figure supplement(s) for figure 6:

**Source data 1.** Quantitative real-time PCR (qRT-PCR) and densitometry analysis on wild-type (WT) and myopalladin knockout (MKO) male mice subjected to transaortic constriction (TAC) or SHAM.

**Figure supplement 1.** Immunofluorescence stainings of left ventricular tissue from the heart of 4-month-old wild-type (WT) and myopalladin knockout (MKO) male mice for palladin (PALLD) and myopalladin (MYPN)-interacting proteins.

PALLD is located in a proline-rich region, not present in MYPN (**Rönty et al., 2005**). Correspondently, MYPN did not bind to SORBS2 in the Y2H system (data not shown).

## CMC contractility and calcium handling

We next determined the effect of MYPN knockout on CMC contractile function and twitch $Ca^{2+}$ transients at different frequencies in ventricular CMCs isolated from 10-week-old MKO mice under

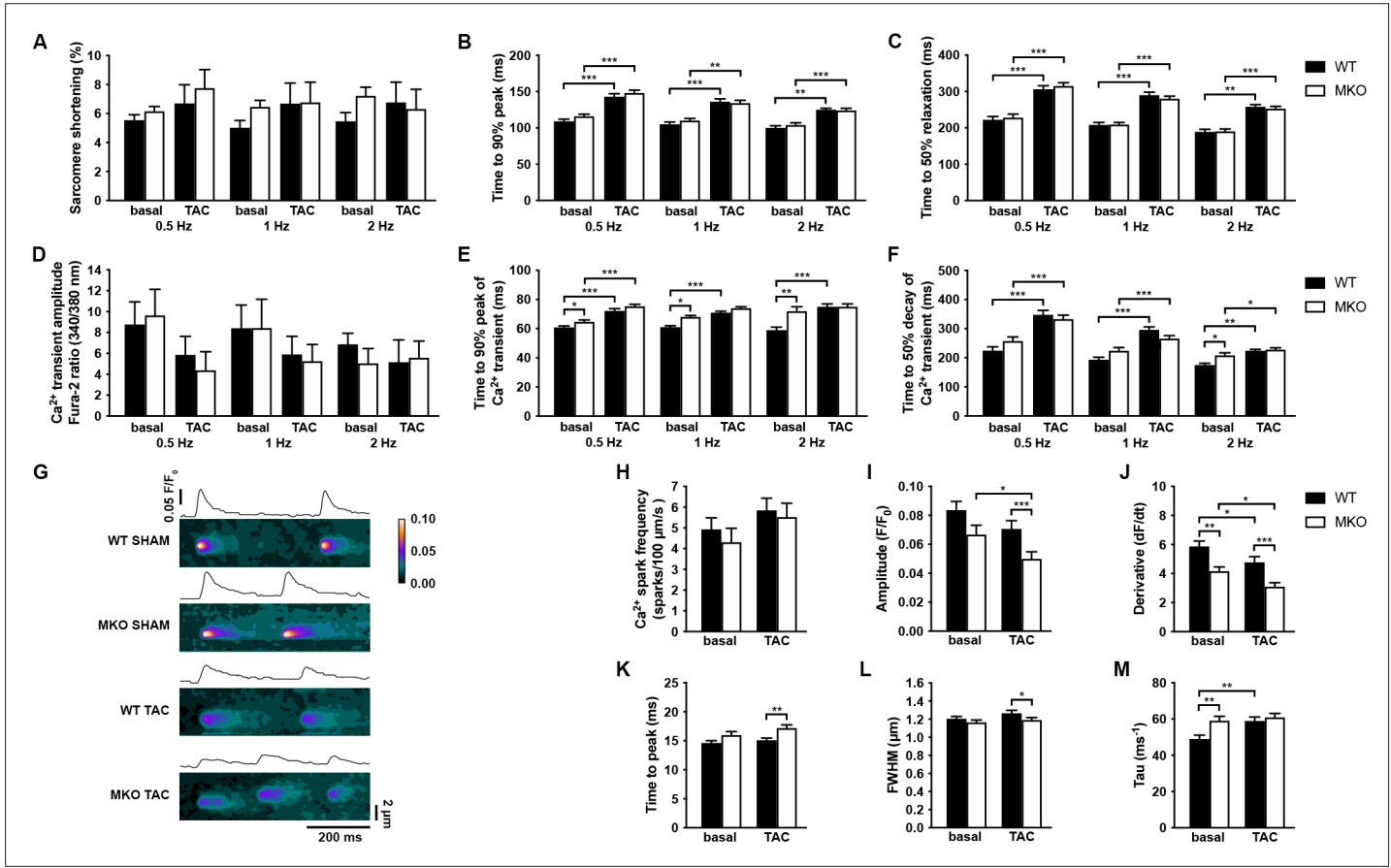

**Figure 7.** Functional analyses on adult cardiomyocytes (CMCs) from wild-type (WT) and myopalladin knockout (MKO) male mice 7 days after transaortic constriction (TAC) or SHAM surgery. (**A–C**) Sarcomere shortening, time to 90 % peak, and time to 50 % relaxation (n = 62 cells from nine WT mice and 71 cells from seven MKO mice). (**D–F**) Amplitude of $Ca^{2+}$ transient, time to 90 % peak of $Ca^{2+}$ transient, and time to 50 % decay of $Ca^{2+}$ transient (n = 29 cells from eight WT mice and 16 cells from six MKO mice). (**G**) Representative $Ca^{2+}$ spark images from WT and MKO male mice subjected to TAC or SHAM for 7 days. (**H–M**) $Ca^{2+}$ spark frequency, amplitude, velocity of $Ca^{2+}$ release (derivative; dF/dt), time to peak, full width at half maximum (FWHM), and tau (n = 355 sparks from 33 cells from three WT SHAM mice, 403 sparks from 27 cells from three MKO SHAM mice, 462 sparks from 31 cells from three WT TAC mice, and 429 sparks from 33 cells from three MKO TAC mice). Data are represented as mean ± standard error of the mean (SEM) as determined by hierarchical analysis (**Sikkel et al., 2017**). *p < 0.05, **p < 0.01, ***p < 0.001; two- or three-level hierarchical testing with Bonferroni correction, as appropriate.

The online version of this article includes the following figure supplement(s) for figure 7:

**Source data 1.** Analysis of sarcomere shortening, $Ca^{2+}$ transients, and $Ca^{2+}$ sparks in cardiomyocytes (CMCs) from wild-type (WT) and myopalladin knockout (MKO) male mice subjected to transaortic constriction (TAC) or SHAM.

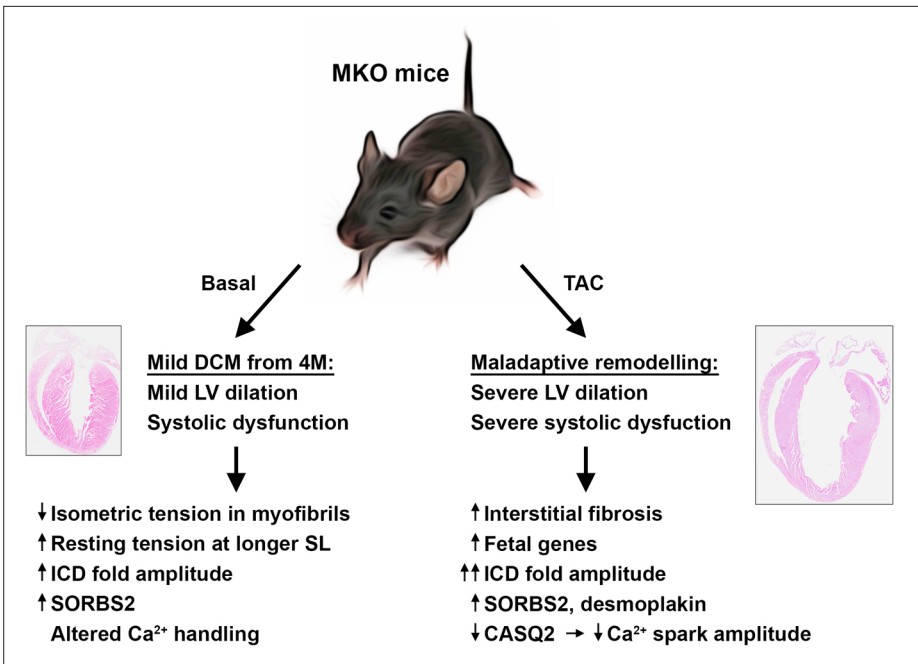

**Figure 8.** Overview figure illustrating the phenotype of myopalladin knockout (MKO) mice under basal conditions and following transaortic constriction (TAC). M, month; LV, left ventricle; SL, sarcomere length, ICD, intercalated disc.

basal conditions and following 7 days of TAC, a time point at which alterations at the CMC level would be expected, but before the development of maladaptive cardiac remodeling in MKO mice (*Figure 7A–F*). No differences in sarcomere shortening were observed between WT and MKO mice, which also showed a similar increase in time to peak (TTP) and time to relaxation after TAC (*Figure 7A–C*). Similarly, TTP and time to decay of the $Ca^{2+}$ transient were prolonged in WT mice after TAC, while in MKO mice the TTP was only significantly increased at 0.5 Hz (*Figure 7D–F*) due to a prolongation of the TTP under basal conditions to levels comparable to those observed after TAC. Additionally, the time to decay was increased in MKO mice at 2 Hz under basal conditions. The $Ca^{2+}$ transient amplitude was similar between WT and MKO mice. These results show prolongation of $Ca^{2+}$ transients in MKO mice under basal conditions.

To determine the consequence of the reduced CASQ2 in MKO mice 4 days after TAC, we analyzed spontaneous $Ca^{2+}$ spark properties (diastolic sarcoplasmic reticulum [SR] $Ca^{2+}$ leak) (*Figure 7G–M*). $Ca^{2+}$ spark frequency was not significantly altered in MKO CMCs either under basal conditions or after TAC (*Figure 7H*). However, while the $Ca^{2+}$ spark decay time constant (tau) was prolonged in WT mice after TAC, it was increased to post-TAC levels in MKO mice under basal conditions (*Figure 7M*). Additionally, the velocity of $Ca^{2+}$ release (derivative; dF/dt) was reduced both under basal conditions and following TAC in MKO mice (*Figure 7J*). On the other hand, prolonged TTP (*Figure 7K*) and reduced $Ca^{2+}$ amplitude (*Figure 7I*) and $Ca^{2+}$ spark width (full width at half maximum, FWHM) (*Figure 7L*) were observed in MKO mice only after TAC, likely as a result of the decreased CASQ2 level in MKO mice compared to WT mice 7 days after TAC.

## Discussion

In the present study, we found that MKO mice develop a relative mild form of progressive DCM, characterized by enlargement of the left ventricle and impaired systolic function without evidence of fibrosis or apoptosis (summarized in *Figure 8*). At the ultrastructural level, MKO mice showed normal sarcomere organization, but a slightly increased ICD fold amplitude. This was associated with decreased myofibrillar isometric tension generation and increased resting tension at longer sarcomere lengths, while force kinetics and calcium sensitivity were unaffected. As MYPN is unlikely to affect the cross-bridge unitary force, the reduction in force generating capacity is likely due to structural

reasons. Since the passive mechanical properties of myofibrils are mostly titin-dependent (*Franssen and González Miqueo, 2016*), we determined titin isoform expression and phosphorylation of known sites affecting stiffness, but did not find evidence of alterations in titin that could explain the increased stiffness. However, it cannot be excluded that other posttranslational changes in titin are affecting stiffness in MKO mice.

In response to biomechanical stress by TAC-induced mechanical pressure overload, MKO mice rapidly developed maladaptive cardiac remodeling, characterized by eccentric hypertrophy with left ventricular dilation and severely impaired systolic function (illustrated in *Figure 8*). This was accompanied by reactivation of the fetal gene program and myocardial fibrosis, hallmarks of pathological remodeling and heart failure. Furthermore, MKO mice showed more convoluted ICDs compared to WT mice, an observation frequently associated with DCM (*Perriard et al., 2003*; *Wilson et al., 2014*). In contrast, WT control mice exhibited compensatory cardiac hypertrophy without a major increase in the left ventricular internal dimension.

We previously demonstrated that MYPN indirectly binds to the titin N2A region in the I-band through its interaction with CARP (*Bang et al., 2001*; *Miller et al., 2003*). In the present study, we discovered that MYPN as well as PALLD directly bind to the titin Ig domains Z4-Z5 at the Z-line. Titin is the largest known protein and stretches from the Z-line to the M-band, acting as a molecular spring responsible for passive force in striated muscle and for keeping the thick filament centered during contraction (reviewed in *Chauveau et al., 2014*; *Linke and Hamdani, 2014*). Titin is essential for sarcomere assembly and organization and plays an important role in mechanosensing and force transmission at the Z-line. The direct as well as indirect binding of MYPN to titin, its dual localization in the sarcomere and the nucleus, and the maladaptive response of MKO mice to biomechanical stress point to a possible role of MYPN in mechano-dependent signaling. However, the function of MYPN in the nucleus is unknown and the exact mechanism by which MYPN would transmit biomechanical stress to the nucleus remains to be discovered. Although ultrastructural studies and immunostainings showed normal sarcomere structure, another, more likely, possibility is that MYPN plays mainly a structural role and that its absence weakens the Z-line, where it binds to α-actinin (*Bang et al., 2001*), nebulette (*Bang et al., 2001*), titin, and PDZ-LIM family members (*von Nandelstadh et al., 2009*). In support of this, we recently found progressive Z-line widening with age and eccentric contraction-induced Z-line abnormalities after downhill running in MKO skeletal muscle (*Filomena et al., 2020*), demonstrating the important role of MYPN for the maintenance of Z-line integrity in skeletal muscle during aging and exercise.

In our recent study of the skeletal muscle phenotype of MKO mice, we found that MKO mice have a reduced myofiber cross-sectional area in skeletal muscle and consequently show decreased isometric force and power output as a result of reduced SRF signaling (*Filomena et al., 2020*). The SRF pathway is regulated by actin dynamics as its activation is dependent on the nuclear translocation of its cofactor MRTF-A, which is sequestered in the cytosol through binding to globular actin and released upon actin polymerization (*Olson and Nordheim, 2010*). We demonstrated that MYPN binds and bundles F-actin as well as interacts with MRTF-A, thereby affecting actin dynamics and consequently the SRF pathway. In contrast to skeletal muscle, the SRF pathway does not appear to be affected in MKO heart as there was no downregulation of known SRF target genes, such as *Nppa*, *Nppb*, *Acta1*, *Actc1*, *Myh6*, *Myh7*, and *Ankrd1* (reported in the Harmonizome database; under 'ENCODE Transcription Factor Targets'; https://amp.pharm.mssm.edu/Harmonizome/gene_set/). In fact, after TAC, *Nppa*, *Nppb*, *Actc1*, *Myh7*, and *Ankrd1*, which are markers of biomechanical stress, were significantly more upregulated in MKO mice compared to WT mice, consistent with the maladaptive response of MKO mice to TAC. Thus, the pathological response of MKO mice to TAC does not appear to be directly related to alterations in the SRF pathway.

Western blot analyses for the expression of MYPN-interacting proteins and activation of cardiac signaling pathways showed no differences between WT and MKO mice. However, analysis for ICD proteins showed increased SORBS2 protein levels in MKO mice both under basal conditions and after TAC as well as upregulation of desmoplakin in MKO mice 4 days after TAC. SORBS2 is an adaptor protein located at the adherens junction of the ICD and the Z-line (*Ding et al., 2020*), where it interacts with a variety of cytoskeletal proteins, including F-actin, α-actinin, and PALLD (*Rönty et al., 2005*; *Sanger et al., 2010*). SORBS2 has been reported to be upregulated in patients with left ventricular noncompaction cardiomyopathy (*Li et al., 2020*) and heart failure (*Vigil-Garcia et al.,*

*2021*). Adeno-associated virus 9-mediated overexpression in mouse heart resulted in microtubule densification, T-tubule disorganization, defective $Ca^{2+}$ handling, and cardiac systolic dysfunction (*Li et al., 2020*). In particular, reduced $Ca^{2+}$ amplitude and prolonged TTP and time to decay of the $Ca^{2+}$ transient were observed in isolated CMCs from SORBS2 overexpressing mice, similar to what we observed in MKO mice, suggesting that the altered $Ca^{2+}$ handling in MKO mice may be at least partly explained by the upregulation of SORBS2. Since both SORBS2 and MYPN are tightly associated with the actin cytoskeleton, it is possible that the upregulation of SORBS2 in MKO mice is related to alterations in the actin cytoskeleton due to the absence of MYPN. Furthermore, increased expression of ICD proteins and higher ICD fold amplitude are commonly associated with the pathology of DCM (*Ortega et al., 2017*; *Perriard et al., 2003*; *Wilson et al., 2014*), so the upregulation of SORBS2 and desmoplakin may be related to the DCM phenotype of MKO mice.

Analysis of proteins involved in calcium homeostasis showed increased protein levels of CASQ2 28 days after TAC in both WT and MKO mice, while CASQ2 levels were reduced in MKO mice compared to WT mice 4 days after TAC. CASQ2 RNA levels were not significantly altered after TAC, suggesting that CASQ2 is regulated at the posttranslational level. Otherwise, there were no changes in the expression or phosphorylation of other calcium-related proteins. CASQ2 is a $Ca^{2+}$-binding protein located in the lumen of the SR, where it acts as a $Ca^{2+}$ buffer and an inhibitory modulator of SR $Ca^{2+}$ release by regulation of RyR2 to prevent premature spontaneous SR $Ca^{2+}$ release during conditions of high SR $Ca^{2+}$ load (*Chopra et al., 2007*; *Knollmann et al., 2006*; *Song et al., 2007*). CASQ2 is a known interaction partner of CARP, although CARP has not been detected in the SR, but it was suggested that CASQ2 is present in the cytoplasm at low levels, allowing for the interaction to take place (*Torrado et al., 2005*). MYPN may thus be indirectly linked to CASQ2 through CARP. However, we consider it most likely that the downregulation of CASQ2 in MKO mice at the early stage after TAC is a secondary effect of the alterations due to MYPN ablation and SORBS2 upregulation. A reduction of CASQ2 levels in CMCs has been reported to result in reduced twitch $Ca^{2+}$ transient amplitude and duration as a result of decreased $Ca^{2+}$ SR content (*Terentyev et al., 2003*). Furthermore, $Ca^{2+}$ spark amplitude and duration were reduced, while $Ca^{2+}$ spark frequency and velocity of $Ca^{2+}$ release were unaffected (*Terentyev et al., 2003*). In contrast, CASQ2 deficiency or reduction in mouse had only minor effects on SR $Ca^{2+}$ content, $Ca^{2+}$ transients, and contractile function under basal conditions (*Chopra et al., 2007*; *Knollmann et al., 2006*; *Song et al., 2007*). However, while diastolic SR $Ca^{2+}$ leak ($Ca^{2+}$ spark frequency) was unaltered at basal levels, isoproterenol stimulation caused increased SR $Ca^{2+}$ leak, making CASQ2-deficient mice more susceptible to stress-induced ventricular arrhythmias. Differences between acute (CMCs) and chronic CASQ2 reduction (knockout mice) may explain the different consequences of CASQ2 modulation in vitro and in vivo. The reduced $Ca^{2+}$ spark amplitude and width as well as increased $Ca^{2+}$ spark TTP in CMCs from MKO mice 7 days after TAC are consistent with the published observations in CMCs with reduced CASQ2 levels (*Terentyev et al., 2003*), and may thus be explained by the reduced CASQ2 levels in MKO mice 4 days after TAC. On the other hand, the alterations under basal conditions, including prolonged $Ca^{2+}$ transient TTP and time to decay as well as increased $Ca^{2+}$ spark time to decay to levels comparable to those of WT mice after TAC are likely unrelated to CASQ2 as CASQ2 levels were unaltered under basal conditions. The same applies for the velocity of $Ca^{2+}$ release from the SR, which was reduced both under basal conditions and after TAC. These changes may instead be related to the upregulation of SORBS2 as discussed above. Delayed $Ca^{2+}$ release and reuptake are typical signs of heart failure (*Roe et al., 2015*) and have been linked to slowed $Ca^{2+}$ spark kinetics due to reorganization of RyR2 clusters (*Kolstad et al., 2018*). Thus, as these alterations happen before the manifestation of cardiac dysfunction in MKO mice, altered $Ca^{2+}$ handling likely contributes to the development of DCM in MKO mice.

Human heterozygous *MYPN* gene mutations have been associated with HCM, DCM, and RCM (*Bagnall et al., 2010*; *Duboscq-Bidot et al., 2008*; *Meyer et al., 2013*; *Purevjav et al., 2012*) and their effect in vivo has been studied in two animal models: transgenic mice overexpressing the MYPN-Y20C variant (allele frequency of 9.34E-4; GnomAD v.2.2.1) linked to DCM and HCM in human (*Purevjav et al., 2012*), and knockin mice carrying the MYPN-Q526X nonsense mutation, equivalent to the human MYPN-Q529X mutation associated with RCM (*Huby et al., 2014*). MYPN-Y20C transgenic mice developed HCM and severely disrupted ICDs, associated with impaired binding of MYPN to CARP, reduced CARP protein levels, and failure of MYPN-Y20C to translocate to the nucleus (*Purevjav et al., 2012*). However, since MYPN-Y20C was overexpressed at high levels,

the physiological significance of these findings is unclear. Heterozygous MYPN-Q529X knockin mice expressed a truncated 65 kDa peptide in the nucleus and developed RCM, characterized by diastolic dysfunction and interstitial and perivascular fibrosis (*Huby et al., 2014*). This was associated with decreased protein levels of CARP as well as reduced phosphorylation levels of Erk1/2, Smad2, and Akt, while protein levels of nebulette, desmin, and MLP/Csrp3 were increased. In contrast, homozygous MYPN-Q526X knockin mice had barely detectable MYPN-Q526X levels as a result of nonsense-mediated mRNA instability and showed no discernable cardiac alterations, although the mice were not studied under conditions of stress (*Huby et al., 2014*). Thus, the cardiomyopathy-associated *MYPN* mutations are thought to have dominant negative effects. Since MKO mice show a relatively mild form of slowly progressive DCM under basal conditions and MYPN-Q526X knockin mice were only studied until 12 weeks of age, it is likely that the phenotype was missed and that homozygous MYPN-Q526X knockin mice, like MKO mice, would develop cardiac systolic dysfunction with age or in response to stress.

Biallelic loss-of-function mutations in MYPN have recently been identified in patients with nemaline myopathy (*Miyatake et al., 2017*), cap myopathy (*Lornage et al., 2017*), and congenital myopathy with hanging big toe (*Merlini et al., 2019*), some of which had mild cardiac involvement. Although no patient developed DCM, this is consistent with the relatively weak cardiac phenotype of MKO mice under basal conditions. The maladaptive response of MKO mice to pressure overload suggests that these patients may be more prone to develop cardiac disease in response to cardiovascular risk factors, such as hypertension, obesity, diabetes, high cholesterol, smoking, etc. Thus, our results highlight the importance of periodic monitoring of heart function in these patients and the need to reduce cardiovascular risk factors by encouraging healthy life style choices and timely therapeutic management.

# Materials and methods
## Plasmid constructs
MYPN, PALLD, and TTN cDNAs were isolated by PCR using available constructs or a human cDNA library as a template and cloned into pGADT7 AD (Takara Bio), pGBKT7 DNA-BD (Takara Bio), pET-3d (Merck Life Science), and pETM-14 (*Dümmler et al., 2005*) vectors as well as a modified pLexA vector (*Stenmark et al., 1995*) using digestion cloning or the In-Fusion HD cloning kit (Takara Bio) according to the manufacturer's instructions. Primer sequences are listed in *Supplementary file 1*. All constructs were confirmed by sequencing.

## Y2H assays
Y2H screening was performed essentially as described previously (*Young et al., 1998*). A region containing titin Ig domains Z4-Z5 (Res. 942–1173; Acc. NM_001256850.1) was cloned into a modified pLexA vector (*Stenmark et al., 1995*) as described in the "Plasmid constructs" section (see *Supplementary file 1* for primer sequences). Briefly, the plasmid was transformed into the *Saccharomyces cerevisiae* L40 reporter strain and subsequently co-transformed with a human skeletal muscle cDNA library in the pGAD10 vector (HL4010AB; Clontech Laboratories). Cells were plated on synthetic defined (SD) agar plates lacking histidine, leucine, and tryptophan (SD/–His/–Leu/–Trp) and incubated at 30°C until colonies appeared (3–4 days). Colonies were verified by colony lift filter assays for β-galactosidase activity after growth on SD/–His/–Leu/–Trp plates containing 40 µg/ml X-α-Gal. Library plasmids from positive clones were isolated and their inserts sequenced. To confirm the interaction and narrow down the binding sites, pGBKT DNA-BD and pGADT7 AD vectors (Takara Bio) containing cDNAs encoding regions of MYPN, PALLD, and TTN were co-transformed into the Y2H Gold yeast strain (Takara Bio). Transfected cells were spotted on SD agar plates lacking tryptophan, leucine, histidine, and adenine (SD/–Ade/–His/–Leu/–Trp), containing 120 ng/ml Aureobasidin A and 40 µg/ml X-α-Gal. Interaction was verified after 3–4 days of incubation at 30°C. Successful transformation of the two plasmids was confirmed by growth on SD/–Trp/–Leu plates. Possible autoactivation of bait and prey constructs was tested by co-transformation of the bait or prey vector with empty prey or bait vector, respectively.

## Protein expression and purification

The human MYPN C-terminal region including three Ig domains (MYPN C-term; Res. 813–1320; Acc. NM_032578.3) and the titin Ig domains Z4-Z5 (Res. 942–1173; Acc. NM_001256850.1) were cloned into the pET-3d expression vector for expression of protein with an N-terminal 6xHis-tag, while the PALLD C-terminal region (PALLD C-term; Res. 794–1123; NM_001166108.1) was cloned into the pETM-14 expression vector for expression of protein with an N-terminal 6xHis-tag followed by a human rhinovirus 3 C protease cleavage site (see *Supplementary file 1*). The plasmids were transformed into the BL21-CodonPlus (DE3)-RIPL strain (Agilent) and heterologous protein expression was obtained by induction with 0.4 mM isopropyl β-D-1 thiogalactopyranoside at 30 °C for 4 hr (MYPN and PALLD C-term) or auto-induction media (*Studier, 2005*) at 30 °C for 48 hr (titin Ig domains Z4-Z5). Cells were harvested by centrifugation and resuspended in lysis buffer containing 20 mM Tris-HCl (pH 8 for titin Ig domains Z4-Z5 and pH 7.2 for MYPN and PALLD C-term), 150 mM NaCl, 4 mM DTT, 2 × Protease Inhibitor Cocktail (Roche), 0.5 mM PMSF, and 20 mM imidazole. Following addition of 0.5 mg/ml lysozyme, cells were incubated for 15 min at 4 °C with shaking and subsequently disrupted by sonication. The lysates were clarified by centrifugation (15,000 rpm for 30 min at 4 °C) and filtration (0.45 μm, Sartorious) before loading onto a pre-equilibrated 2 × 1 ml HisTrap column (GE Healthcare). The column was washed with 10 volumes of lysis buffer, whereafter bound proteins were eluted with 500 mM imidazole. Further purification was performed by size exclusion chromatography on a HiLoad Superdex 75 column (GE Healthcare) equilibrated with 20 mM Tris-HCl (pH dependent on the protein), 150 mM NaCl, 5 mM EDTA, and 4 mM DTT. Glycerol was added at a final concentration of 10% for storage.

## MST assay

MST (*Seidel et al., 2013*) was performed using a Monolith NT.115 instrument (NanoTemper Technologies). MYPN C-terminal and titin Ig domain Z4-Z5 peptides were labeled with NT-647 dye via NHS-driven amino coupling using the Monolith NT Protein Labeling Kit RED-NHS (NanoTemper Technologies) according to the manufacturer's instructions. UV-visible spectroscopy was used for assessment of labeling ratio (determined as the 280/650 nm ratio) and protein concentration (280 nm). Optimal substrate concentrations for the binding assay were determined on the instrument with 20% LED power in capillaries of different matrix. Hydrophobic capillaries and 400 nM MYPN C-term or 450 nM titin Ig domains Z4-Z5 were found to give the sharpest signal output and were used as fixed parameters in the binding experiments. Titration of interacting partners at different concentrations was performed after incubation of mixed protein solutions for 30 min at room temperature in 20 mM Tris-HCl pH 8, 150 mM NaCl, 5 mM EDTA, and 4 mM DTT, after which thermal gradients were generated using an MST power of 40% and 80%. Raw data were analyzed using NT Affinity Analysis software (v2.0.1334) (NanoTemper Technologies). Experiments showing Gaussian distribution of capillary fluorescence and equal loading of labeled protein were averaged and fitted to the Hill equation: $Y = B_{max}*X^h/(K_d^h + X^h)$, using Prism 9 (GraphPad) software to estimate the equilibrium dissociation constant ($k_d$).

## Animal experiments

All animal experiments were approved by the Italian Ministry of Health and performed in full compliance with the rules and regulations of the European Union (Directive 2010/63/EU of the European Parliament) and Italy (Council of September 22, 2010; directive from the Italian Ministry of Health) on the protection of animals used for scientific purposes. Mice used for experiments were sacrificed by cervical dislocation under isoflurane anesthesia. Animals were randomly assigned to different experimental groups before the start of experiments. The investigators were blinded to genotype and treatment.

## In vivo cardiac physiology

The generation of MKO mice has recently been described (*Filomena et al., 2020*). The mice were backcrossed for at least 10 generations in the C57BL/6J background. Mice anesthetized with 1% isoflurane were subjected to transthoracic echocardiography using a Vevo 2100 System (VisualSonics) and a 30 MHz probe as previously described (*Tanaka et al., 1996*). For surgical procedures, mice were anesthetized by intraperitoneal injection of a mixture of ketamine (100 mg/kg) and xylazine

(5 mg/kg) and the depth of anesthesia was monitored by toe pinch. Buprenorphine (0.02 mg/kg) was administered for postoperative analgesia during the first 48 hr after surgery. TAC was performed with a 27-gauge needle on 10-week-old WT and MKO mice as previously described (*Tanaka et al., 1996*). Cardiac morphology and function were assessed by transthoracic echocardiography and the gradient for the arterial blood pressure between the constriction was evaluated by Doppler echocardiography. Only mice with a pressure gradient >70 mmHg were included in the analysis. SHAM-operated mice were used as controls.

## Isolation of adult ventricular CMCs

For isolation of ventricular CMCs, hearts were cannulated and mounted on a Langendorff perfusion apparatus, and perfused with perfusion buffer composed of Hank's Balanced Salt Solution (HBSS w/o $CaCl_2$ and $MgCl_2$; Life Technologies) supplemented with 1.2 mM $MgSO_4$, 15 mM glucose, 30 mM taurine, and 1 mM $MgCl_2$ hexahydrate for 20 min at 37°C. Collagenase Type 2 (2.2 mg/ml; Worthington Biochemical Corporation) was then added to the solution and perfusion was continued for about 10 min until the heart became flaccid. Subsequently, the heart was removed and cells were dissociated and filtered through a 70 µm filter after which bovine serum albumin (fraction V; Merck Life Science) was added to a final concentration of 4% to inactivate the enzyme. The cells were allowed to settle and resuspended in fresh solution. For determination of CMC size, pictures were taken on an Olympus BX51 Fluorescent microscope and analyzed using ImageJ, version 2.1.0/1.53 C (NIH).

## CMC contractility and intracellular $Ca^{2+}$ transient measurements

Simultaneous measurements of CMC contractility and $Ca^{2+}$ transients were carried out on an IonOptix system as previously described (*Kondo et al., 2006*). Briefly, CMCs loaded with 1 µM of the $Ca^{2+}$ probe Fura-2 AM (Thermo Fisher Scientific) were placed in a perfusion system and continuously perfused with perfusion buffer (HBSS without $Ca^{2+}$ and $Mg^{2+}$, supplemented with 1.2 mM $MgSO_4$, 15 mM glucose, 30 mM taurine, and 1.0 mM $MgCl_2$), containing 1.0 mM $CaCl_2$ at 37°C. Loaded cells were paced at 0.5, 1.0, and 2.0 Hz, and sarcomere shortening and Fura-2 ratio (measured at 512 nm upon excitation at 340 and 380 nm) were simultaneously recorded on a Nikon Eclipse TE-2000S inverted fluorescence microscope with a 40×/1.3 NA objective and an attached CCD camera (MyoCam-S, IonOptix). Data acquisition and analysis were performed using Ion Wizard software, version 6.6.11 (IonOptix).

## Spontaneous $Ca^{2+}$ spark investigation

For SR $Ca^{2+}$ spark measurements, CMCs were loaded with 10 µM of Fluo-4 AM (Thermo Fisher Scientific) as previously described (*Lyon et al., 2009*). Acquisition was performed using a two-photon microscope (TriM Scope II, LaVision BioTec) during continuous perfusion with temperature-controlled perfusion buffer containing 1.0 mM $Ca^{2+}$ at 37°C. Fluo-4 was excited at 810 nm and the fluorescence emission was collected with a 525/50 filter. Images were acquired in *xt* line scan mode and event detection was performed using the SparkMaster plugin (*Picht et al., 2007*) of ImageJ (NIH) with a detection criteria of 3.8. The following $Ca^{2+}$ spark parameters were determined: amplitude ($F/F_0$), frequency, derivative (dF/dt), FWHM, full duration at half maximum (FDHM), TTP, and the tau constant of decay.

## Mechanical experiments in isolated myofibrils

Mechanical data were collected at 15°C from small bundles of cardiac myofibrils from frozen ventricular strips of WT and MKO mice as previously described (*Kreutziger et al., 2011*). Briefly, thin myofibril bundles (1–4 µm width, initial sarcomere length around 2.2 µm) were maximally $Ca^{2+}$-activated (pCa 4.5) and fully relaxed (pCa 8.0) by fast solution switching. Maximal tension and the kinetics of force activation and force relaxation were measured. Quasi-steady-state sarcomere-length-resting tension relations were determined in relaxing solution (pCa 8.0) for both myofibril groups as previously described (*Mastrototaro et al., 2015*). In a set of experiments, the tension generated by WT and MKO myofibrils at maximal and submaximal $Ca^{2+}$ activation were compared using $Ca^{2+}$ jump protocols to investigate whether the lack of MYPN affects myofilament $Ca^{2+}$ sensitivity. Both channels of the perfusing pipette were loaded with $Ca^{2+}$-activating solutions, one with pCa 4.5 activating solution (maximal activation) and the other with pCa 5.75 activating solution (submaximal activation). Myofibrils were activated by translating the interface between the relaxing solution in the experimental

chamber and the flow of one of the two activating solutions. Once a steady plateau of isometric force was attained, the perfusion flow was rapidly switched to the second activating solution and the ratio between the tension levels generated at submaximal and maximal activation was calculated. The sequence of the tested solutions was alternated between preparations.

## Histology and immunofluorescence stainings

For histology, mouse hearts were harvested, relaxed in 50 mM KCl in phosphate-buffered saline (PBS), and fixed overnight in 4% paraformaldehyde (PFA) in PBS. Subsequently, hearts were dehydrated, embedded in paraffin, and cut in 8 µm sections in the four-chamber view. Briefly, heart sections were stained with H&E or Picro Sirius Red and imaged using a VS20 DotSlide Digital Virtual Microscopy System (Olympus). The area of fibrosis in the left ventricle was determined using ad hoc software automatically detecting Picro Sirius Red-stained areas based on RGB color segmentation (*Grizzi et al., 2019*). The sum of Picro Sirius Red-stained areas was expressed as a percentage of the left ventricular area excluding unfilled and tissue-free spaces. For quantification of CMC size, deparaffinized paraffin sections were UV-treated for 10 min after which they were incubated with WGA Alexa Fluor 594 Conjugate (Thermo Fisher Scientific, 1:500) at 4°C overnight. Sections were mounted with VECTASHIELD Vibrance Antifade Mounting Medium with DAPI (D.B.A. Italia Srl.) and imaged using a Leica DMi8 widefield fluorescent microscope. CMC size of transversely cut CMCs was measured using ImageJ (NIH). For immunostainings, the upper part of the heart was relaxed in 50 mM KCl in PBS, fixed for 15 min in 4% PFA in PBS, and subsequently saturated in 15% and 30% sucrose in PBS and frozen in OCT; 10 µm sections were permeabilized and blocked for 1 hr in blocking solution containing 5% normal goat serum, 0.3% Triton X-100, and 50 mM glycine in PBS after which sections were incubated with primary antibodies in wash buffer (blocking buffer diluted 10 times in PBS) overnight at 4°C. The following primary antibodies were used: PALLD (*Pogue-Geile et al., 2006*) (1:30, kindly provided by Prof. Carol Otey, University of North Carolina, Chapel Hill, NC), ANKRD1/CARP (1:20) (*Miller et al., 2003*), α-actinin (1:250; Merck Life Science #A7811), cypher (*Zhou et al., 2001*) (1:50, kindly provided by Prof. Ju Chen, University of California San Diego, La Jolla, CA), desmin (1:80; Abcam #Ab8592), and MKL1/MRTF-A (1:30; Merck Life Science #AV37504). After washing, sections were incubated at room temperature for 4 hr with rhodamine-labeled phalloidin (1:100, Thermo Fisher Scientific) or Alexa-Fluor-488, -568, or -647-conjugated IgG secondary antibodies (1:500, Thermo Fisher Scientific) and mounted with VECTASHIELD Vibrance Antifade Mounting Medium with DAPI (D.B.A. Italia Srl.). Confocal microscopy was performed on a Leica SP8 inverted confocal microscope with a 60× oil immersion lens. Individual images (1024 × 1024) were converted to tiff format and merged as pseudocolor RGB images using ImageJ (NIH).

## Transmission electron microscopy

For transmission electron microscopy, the heart was excised and fixed in 2% PFA and 2% glutaraldehyde in 0.15 M sodium cacodylate buffer, pH 7.4 as previously described (*Zhang et al., 2008*). Briefly, the left ventricle was cut into small pieces (~1 mm cubes), postfixed with a mixture of 2% osmium tetroxide and 3% potassium ferrocyanide for 1 hr, and stained overnight in 0.5% uranyl acetate at 4°C. The following day, tissue was stained for 2 hr in 2% uranyl acetate and subsequently dehydrated in a series of ethanol and propylene oxide before embedding with the epoxy resin (EMbed-812). Ultra-thin sections (60–70 nm) were cut with an Leica EM UC7 ultramicrotome (Leica Microsystems, Wetzlar, Germany). Electron micrographs were acquired using an FEI Tecnai G2 Spirit BioTWIN electron microscope equipped with a Veletta CCD camera (FEI Europe B.V., Eindhoven, The Netherlands). Analysis of ICD fold amplitude was performed as previously described by defining the ICD fold amplitude as the distance between the transitional junctions defining the beginning of the ICD fold (*Wilson et al., 2014*). More specifically, ICD fold amplitude was measured in longitudinal regions with at least two sarcomeres on each side of the ICD as $(L_2 - L_1)/2$, where $L_1$ was defined as the distance corresponding to two sarcomere lengths plus the ICD and $L_2$ as the distance corresponding to four sarcomere lengths plus the ICD. When possible, the ICD fold amplitude was measured at several locations for each ICD, whereafter the average was calculated.

## RNA extraction and (qRT-PCR)

Total RNA was isolated from left ventricular tissue using TRIzol reagent (Thermo Fisher Scientific) according to the manufacturer's instructions. For qRT-PCR, first-strand cDNA synthesis was performed using the High Capacity cDNA Reverse Transcription kit (Thermo Fisher Scientific), whereafter qRT-PCR was performed in triplicate with custom-designed oligos (see *Supplementary file 1*) using the GoTaq qPCR Master Mix (Promega). Relative expression analysis was performed using the ΔΔCt method using *B2m* for normalization, as it was the most stably expressed gene.

## SDS-PAGE and Western blot analysis

For Western blot analysis, left ventricular tissue was homogenized in RIPA buffer containing 50 mM Tris-HCl pH 7.5, 150 mM NaCl, 0.5 mM dithiothreitol (DTT), 1 mM EDTA, 1% (v/v) sodium dodecyl sulfate (SDS), 1% (v/v) Triton X-100, protease inhibitors (1 mM phenylmethylsulfonyl fluoride [PMSF] and COmplete Protease Inhibitor Cocktail Tablets; Merck Life Science), and Pierce Phosphatase Inhibitor Mini Tablets (Thermo Fisher Scientific) using a TissueLyser II (Qiagen). Protein concentration was determined using the DC Protein Assay Kit II (Bio-Rad Laboratories) according to the manufacturer's instructions. SDS-PAGE was performed using TGX Stain-Free gels (Bio-Rad Laboratories) after which proteins were transferred to PVDF membranes and Western blot analyses were performed using the following primary antibodies: MYPN (1:1000) (*Yamamoto et al., 2013*), PALLD 621 (1:500) (*Pogue-Geile et al., 2006*), NEBL (1:500) (*Mastrototaro et al., 2015*), ANKRD1/CARP (1:200) (*Miller et al., 2003*), cypher (1:500) (*Zhou et al., 2001*), α-actinin (1:50000; Merck Life Science #A7811), MKL-1/MRTF-A (1:500; Immunological Sciences #AB-84312), desmoplakin (1:750; Bio-Rad Laboratories #2722–5204), α-E-catenin (1:1000; Santa Cruz Biotechnology #sc-9988), β-catenin (1:1000; Cell Signaling Technology #8480), γ-catenin (1:1000; Immunological Sciences #AB-90215), vinculin (1:2000; Merck Life Science #V9264), N-cadherin (1:1000; Cell Signaling Technology #4061), SORBS2 (1:1000; Merck Life Science #SAB4200183), connexin 43 (1:400; Thermo Fisher Scientific #35–5000), pAkt-Ser473 (1:500; Immunological Sciences #MAB-94111), Akt (1:1000; Cell Signaling Technology #9272), pErk1/2-Thr202/Tyr204 (1:500; Cell Signaling Technology #4370), Erk1/2 (1:1000; Santa Cruz Biotechnology #sc-514302), pP38-Tyr182 (1:500; Santa Cruz Biotechnology #sc-166182), P38α/β (1:500; Santa Cruz Biotechnology #7972), calcineurin B (1:500; Novus Biochemical #NB-300–728), calsequestrin-2 (1:5000: BD Transduction Labs #C16420), SERCA2A (1:1000; Affinity Bio Reagent [ABR] #ma3-919), pPLN-Thr16 (1:500; Badrilla #010–12), PLN (1:500; Novus Biochemical #NB300-582), RyR2 (1:1000; Affinity Bio Reagent [ABR] #ma3-916), pRyR-Ser2808 (1:500; Badrilla #A010-30AP), sodium calcium exchanger (NCX) (1: 1000; Merck Life Science #N216), CACNA1C (1:500; Abcam #Ab58552), CACNA1S (1:500; Abcam #Ab2862), and actin (1:1000; Santa Cruz Biotechnology #sc-1615). The following HRP-linked secondary antibodies were used: goat anti-rabbit IgG Horseradish Peroxidase (HRP) (1:5000; Thermo Fisher Scientific #31460) and goat anti-mouse IgG-HRP (1:5000; Thermo Fisher Scientific #31430). The Immobilon Western Chemiluminescent HRP Substrate (Merck Life Science) was used and chemiluminescence was detected on a Chemidoc MP System (Bio-Rad Laboratories). Relative protein expression was determined by densitometry using Image Lab, version 5.2.1 software (Bio-Rad Laboratories). Normalization was performed to total protein content as determined on UV-activated TGX Stain-Free gels.

## Titin isoform separation and phospho-titin analysis by Western blot analysis

Separation of titin isoforms by SDS-PAGE and Western blot analyses were performed as previously described (*Hamdani et al., 2013*). Briefly, left ventricular tissue was solubilized in 50 mM Tris-SDS buffer (pH 6.8) containing 8 µg/ml leupeptin (Merck Life Science) and phosphatase inhibitor cocktail (PIC [P2880], 10 µl/ml; Merck Life Science). SDS-PAGE was performed on 1.8% polyacrylamide/1% agarose gels run at 5 mA for 16 hr, whereafter titin bands were visualized by Coomassie blue staining. Titin isoform ratio was calculated as the densitometric value for titin N2BA over N2B. For determination of total phosphoserine/threonine phosphorylation and site-specific titin phosphorylation, Western blots were performed using anti-phosphoserine/threonine antibody (ECM Biosciences #PP2551; 1:500) as well as titin phosphosite-specific antibodies against pTTN-Ser3991 (N2B region; phosphorylated by PKA and ERK2; 1:500), pTTN-Ser4080 (N2B region; phosphorylated by PKG; 1:500), and pTTN-Ser12742 (PEVK region; phosphorylated by PKCα; 1:500) (*Kötter et al., 2013*). Densitometry

was performed by normalization to total protein content as determined by Coomassie blue staining of each blot.

## Statistical analysis

Sample sizes with adequate power to detect statistical differences between groups were determined based on our previous experience and gold standards in the field. The only exclusion criteria were technical failure or death/injury. Data are represented as mean ± standard error of the mean (SEM). Statistical comparisons between WT and MKO mice were done using the unpaired Student's t-test. Simultaneous effects of genotype and another experimental variable were determined using two-way analysis of variance (ANOVA). The Shapiro-Wilk test was performed to confirm normal distribution in each group and in residuals from a linear regression model, Bartlett's test to confirm homogeneity of variance across groups, and Spearman's rank correlation test to confirm heteroscedasticity of residuals. When necessary, data were log-transformed to meet ANOVA assumptions. For the statistical analysis of echocardiographic parameters in WT and MKO mice before and after TAC in *Figure 4*, a linear mixed model was used. Bonferroni correction was used for multiple pairwise comparisons. The functional comparisons of sarcomere shortening and $Ca^{2+}$ transients in CMCs were performed using two-level hierarchical testing with Bonferroni correction, while $Ca^{2+}$ spark parameters were compared using three-level hierarchical testing with Bonferroni correction (*Sikkel et al., 2017*). This was done to eliminate the effects of variations both within cells and between mice. $p < 0.05$ was considered significant. Statistical analysis was performed using Prism 9 (GraphPad) or R (*R Development Core Team, 2021*) software.

## Acknowledgements

We thank Dr Ju Chen (University of California San Diego, CA, USA) in whose lab the initial studies were performed, the Unit of Advanced Optical Microscopy at Humanitas Research Hospital for assistance with microscopy analyses, the Humanitas Histology Unit for assistance with analysis of Picro Sirius Red-stained sections, and Marion von Frieling-Salewsky for performing titin gel electrophoresis and Western blot analysis.

## Additional information

### Competing interests

The other authors declare that no competing interests exist.

### Funding

| Funder | Grant reference number | Author |
| --- | --- | --- |
| Fondazione Telethon | GGP12282 | Marie-Louise Bang |
| Ministero dell'Istruzione, dell'Università e della Ricerca | 2010R8JK2X_006 | Marie-Louise Bang |
| Ministero della Salute | RF-MUL-2007-666195 | Marie-Louise Bang |
| Fondazione Cariplo | 2007.5812 | Marie-Louise Bang |
| Agenzia Spaziale Italiana | 2015-009-R.0 | Marie-Louise Bang |
| European Commission | 777204 | Corrado Poggesi |
| Wellcome Trust | 201543/Z/16 | Mathias Gautel |

The funders had no role in study design, data collection and interpretation, or the decision to submit the work for publication.

### Author contributions

Maria Carmela Filomena, Formal analysis, Investigation, Writing – original draft, Writing – review and editing; Daniel L Yamamoto, Pierluigi Carullo, Nicoletta Piroddi, Beatrice Scellini, Chiara Tesi, Formal

analysis, Investigation; Roman Medvedev, Formal analysis, Methodology, Writing – original draft; Andrea Ghisleni, Formal analysis, Investigation, Writing – original draft; Roberta Crispino, Methodology; Francesca D'Autilia, Jianlin Zhang, Arianna Felicetta, Simona Nemska, Investigation; Simone Serio, Formal analysis; Daniele Catalucci, Formal analysis, Writing – review and editing; Wolfgang A Linke, Supervision, Writing – original draft; Roman Polishchuk, Methodology, Supervision, Writing – review and editing; Corrado Poggesi, Resources, Supervision, Writing – original draft, Writing – review and editing; Mathias Gautel, Supervision, Writing – original draft, Writing – review and editing; Marie-Louise Bang, Conceptualization, Formal analysis, Funding acquisition, Investigation, Methodology, Project administration, Resources, Supervision, Writing – original draft, Writing – review and editing

### Author ORCIDs
Roman Medvedev  http://orcid.org/0000-0003-1174-2993
Andrea Ghisleni  http://orcid.org/0000-0001-8456-5903
Daniele Catalucci  http://orcid.org/0000-0001-7041-6114
Wolfgang A Linke  http://orcid.org/0000-0003-0801-3773
Roman Polishchuk  http://orcid.org/0000-0002-7698-1955
Marie-Louise Bang  http://orcid.org/0000-0001-8859-5034

### Ethics
All animal studies were approved by the Italian Ministry of Health and performed in full compliance with the rules and regulations of the European Union (Directive 2010/63/EU of the European Parliament) and Italy (Council of 22 September 2010; directive from the Italian Ministry of Health) on the protection of animals used for scientific purposes.

### Decision letter and Author response
Decision letter https://doi.org/10.7554/eLife.58313.sa1
Author response https://doi.org/10.7554/eLife.58313.sa2

## Additional files

### Supplementary files
- Supplementary file 1. Oligos used for quantitative real-time PCR (qRT-PCR) and clonings.
- Transparent reporting form

### Data availability
All data generated or analysed during this study are included in the manuscript and supporting files. Source data files have been provided for all figures.

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

# Appendix 1

### Appendix 1—key resources table

| Reagent type (species) or resource | Designation | Source or reference | Identifiers | Additional information |
|---|---|---|---|---|
| Strain, strain background (male *Mus musculus*) | MYPN knockout (MKO) mice (in C57BL/6 J background) | *Filomena et al., 2020* | N/A | |
| Strain, strain background (*Mus musculus*) | C57BL/6 J | The Jackson Laboratories | Cat# 000664 RRID:IMSR_JAX:000664 | |
| Strain, strain background (*Escherichia coli*) | DH5α electrocompetent cells | New England BioLabs | Cat# C2989K | |
| Strain, strain background (*Escherichia coli*) | BL21-CodonPlus (DE3)-RIPL strain | Agilent Technologies | Cat# 230280 | |
| Strain, strain background (*Saccharomyces cerevisiae*) | Y2H Gold yeast strain | Takara Bio | Cat# 630498 | |
| Strain, strain background (*Saccharomyces cerevisiae*) | L40 yeast strain | Takara Bio | N/A | |
| Antibody | Anti-MYPN (rabbit polyclonal) | *Yamamoto et al., 2013* | N/A | WB (1:1000) |
| Antibody | Anti-PALLD 621 (rabbit polyclonal) | *Pogue-Geile et al., 2006* | Kindly provided by Prof. Carol Otey, University of North Carolina, Chapel Hill, NC, USA | WB (1:500) IF (1:30) |
| Antibody | Anti-NEBL (rabbit polyclonal) | *Mastrototaro et al., 2015* | N/A | WB (1:500) |
| Antibody | Anti-ANKRD1/CARP (rabbit polyclonal) | *Miller et al., 2003* | N/A | WB (1:200) IF (1:20) |
| Antibody | Anti-Cypher (rabbit polyclonal) | *Zhou et al., 2001* | Kindly provided by Prof. Ju Chen, University of California San Diego, La Jolla, CA, USA | WB (1:500) IF (1:50) |
| Antibody | Anti-α-actinin (mouse monoclonal) | Merck Life Science | Cat# A7811 RRID:AB_476766 | WB (1:50000) IF (1:250) |
| Antibody | Anti-desmin (rabbit polyclonal) | Abcam | Cat# Ab8592 RRID:AB_306653 | IF (1:80) |
| Antibody | Anti-MKL1/MRTF-A (rabbit polyclonal) | Merck Life Science | Cat# AV37504 RRID:AB_1853972 | IF (1:30) |
| Antibody | Anti-MKL1/MRTF-A (rabbit polyclonal) | Immunological Sciences | Cat# AB-84312 RRID:AB_2892156 | WB (1:500) |
| Antibody | Anti-desmoplakin 1/2 (mouse monoclonal) | Bio-Rad Laboratories | Cat# 2722–5204 RRID:AB_619950 | WB (1:750) |
| Antibody | Anti-α-E-catenin (mouse monoclonal) | Santa Cruz Biotechnology | Cat# sc-9988 RRID:AB_626805 | WB (1:1000) |
| Antibody | Anti-β-catenin (rabbit monoclonal) | Cell Signaling Technology | Cat# 8480 RRID:AB_11127855 | WB (1:1000) |

*Appendix 1 Continued on next page*

*Appendix 1 Continued*

| Reagent type (species) or resource | Designation | Source or reference | Identifiers | Additional information |
|---|---|---|---|---|
| Antibody | Anti-γ-catenin (rabbit polyclonal) | Immunological Sciences | AB-90215 RRID:AB_2892157 | WB (1:1000) |
| Antibody | Anti-vinculin (mouse monoclonal) | Merck Life Science | Cat# V9264 RRID:AB_10603627 | WB (1:2000) |
| Antibody | Anti-N-cadherin (mouse monoclonal) | Cell Signaling Technology | Cat# 4061 RRID:AB_10694647 | WB (1:1000) |
| Antibody | Anti-SORBS2 (mouse monoclonal) | Merck Life Science | Cat# SAB4200183 RRID:AB_10638778 | WB (1:750) |
| Antibody | Anti-connexin 43 (mouse monoclonal) | Thermo Fisher Scientific | Cat# 35–5000 RRID:AB_87322 | WB (1:400) |
| Antibody | Anti-pAkt-Ser473 (rabbit monoclonal) | Immunological Sciences | MAB-94111 RRID:AB_2892158 | WB (1:500) |
| Antibody | Anti-Akt (rabbit polyclonal) | Cell Signaling Technology | Cat# 9272 RRID:AB_329827 | WB (1:1000) |
| Antibody | Anti-pErk1/2-Thr202/Tyr204 (rabbit polyclonal) | Cell Signaling Technology | Cat# 4370 RRID:AB_2315112 | WB (1:500) |
| Antibody | Anti-Erk1/2 (mouse monoclonal) | Santa Cruz Biotechnology | Cat# sc-514302 RRID:AB_2571739 | WB (1:1000) |
| Antibody | Anti-pP38-Tyr182 (mouse monoclonal) | Santa Cruz Biotechnology | Cat# sc-166182 RRID:AB_2141746 | WB (1:500) |
| Antibody | Anti-P38α/β (mouse monoclonal) | Santa Cruz Biotechnology | Cat# sc-7972 RRID:AB_628079 | WB (1:500) |
| Antibody | Anti-calcineurin B (rabbit polyclonal) | Novus Biochemical | Cat# NBP1-32720 RRID:AB_2168483 | WB (1:500) |
| Antibody | Anti-calsequestrin - (mouse monoclonal) | BD Trasduction Labs | Cat# C16420 | WB (1:5000) |
| Antibody | Anti-SERCA2A (mouse monoclonal) | Affinity Bio Reagent (ABR) | Cat# ma3-919 RRID:AB_325502 | WB (1:1000) |
| Antibody | Anti-pPLN-Thr16 (rabbit polyclonal) | Badrilla | Cat# 010–12 RRID:AB_2617047 | WB (1:500) |
| Antibody | Anti-PLN (mouse monoclonal) | Novus Biochemical | Cat# NB300-582 RRID:AB_10000946 | WB (1:500) |
| Antibody | Anti-RyR2 (mouse monoclonal) | Affinity Bio Reagent (ABR) | Cat# ma3-916 RRID:AB_2183054 | WB (1:1000) |
| Antibody | Anti-pRyR-Ser2808 (rabbit polyclonal) | Badrilla | Cat# A010-30AP RRID:AB_2617052 | WB (1:500) |
| Antibody | Anti-sodium calcium exchanger (NCX) (mouse monoclonal) | Merck Life Science | Cat# N216 RRID:AB_260750 | WB (1:1000) |
| Antibody | Anti-CACNA1C (rabbit polyclonal) | Abcam | Cat# Ab58552 RRID:AB_879800 | WB (1:500) |
| Antibody | Anti-CACNA1S (mouse monoclonal) | Abcam | Cat# Ab2862 RRID:AB_2069567 | WB (1:500) |
| Antibody | Anti-phosphoserine/threonine (rabbit polyclonal) | ECM Biosciences | Cat# PP2551 RRID:AB_1184778 | WB (1:500) |
| Antibody | Anti-pTTN-Ser3991 (rabbit polyclonal) | *Kötter et al., 2013* | N/A | WB (1:500) |

*Appendix 1 Continued*

| Reagent type (species) or resource | Designation | Source or reference | Identifiers | Additional information |
|---|---|---|---|---|
| Antibody | Anti-pTTN-Ser4080 (rabbit polyclonal) | *Kötter et al., 2013* | N/A | WB (1:500) |
| Antibody | Anti-pTTN-Ser12742 (rabbit polyclonal) | *Kötter et al., 2013* | N/A | WB (1:500) |
| Antibody | Anti-actin (goat polyclonal) | Santa Cruz Biotechnology | Cat# sc-1615 RRID:AB_630835 | WB (1:10000) |
| Antibody | Goat anti-mouse IgG (H + L) secondary antibody Alexa Fluor 488-conjugated IgG | Thermo Fisher Scientific | Cat# A11029 RRID:AB_138404 | IF (1:500) |
| Antibody | Goat anti-rabbit IgG (H + L) secondary antibody Alexa Fluor 488-conjugated IgG | Thermo Fisher Scientific | Cat# A11034 RRID:AB_2576217 | IF (1:500) |
| Antibody | Goat anti-mouse IgG (H + L) Secondary antibody Alexa Fluor 568-conjugated conjugate | Thermo Fisher Scientific | Cat# A11031 RRID:AB_144696 | IF (1:500) |
| Antibody | Goat anti-rabbit IgG (H + L) Secondary antibody Alexa Fluor 568-conjugated conjugate | Thermo Fisher Scientific | Cat# A11036 RRID:AB_10563566 | IF (1:500) |
| Antibody | Goat anti-mouse IgG (H + L) Secondary antibody Alexa Fluor 647-conjugated conjugate | Thermo Fisher Scientific | Cat# A21236 RRID:AB_2535805 | IF (1:500) |
| Antibody | Goat anti-rabbit IgG (H + L) Secondary antibody Alexa Fluor 647-conjugated conjugate | Thermo Fisher Scientific | Cat# A21245 RRID:AB_2535813 | IF (1:500) |
| Antibody | Goat anti-rabbit IgG Horseradish Peroxidase (HRP) | Thermo Fisher Scientific | Cat# 31460 RRID:AB_228341 | WB (1:5000) |
| Antibody | Goat anti-mouse IgG-HRP | Thermo Fisher Scientific | Cat# 31430 RRID:AB_228307 | WB (1:5000) |
| Antibody | Donkey anti-goat IgG-HRP | Santa Cruz Biotechnology | Cat# sc-2020 RRID:AB_631728 | WB (1:2000) |
| Recombinant DNA reagent | Modified pLexA vector | *Stenmark et al., 1995* | N/A | |
| Recombinant DNA reagent | pGBKT7 DNA-BD vector | Takara Bio | Cat# 630443 | |
| Recombinant DNA reagent | pGADT7 AD vector | Takara Bio | Cat# 630442 | |
| Recombinant DNA reagent | Human skeletal muscle cDNA library in the pGAD10 vector | Clontech Laboratories | HL4010AB | |
| Recombinant DNA reagent | pET-3d vector | Merck Life Science | Cat# 69421 | |

*Appendix 1 Continued on next page*

*Appendix 1 Continued*

| Reagent type (species) or resource | Designation | Source or reference | Identifiers | Additional information |
|---|---|---|---|---|
| Recombinant DNA reagent | pETM-14 | *Dümmler et al., 2005* | N/A | |
| Recombinant DNA reagent | pET-3d 6xHis human MYPN C-term | This paper | N/A | Cloning primers in *Supplementary file 1* |
| Recombinant DNA reagent | pETM-14 human MYPN C-term | This paper | N/A | Cloning primers in *Supplementary file 1* |
| Recombinant DNA reagent | pET-3d-6xHis human Titin Z4-Z5 | This paper | N/A | Cloning primers in *Supplementary file 1* |
| Recombinant DNA reagent | pETM-14 human PALLD C-term | This paper | N/A | Cloning primers in *Supplementary file 1* |
| Recombinant DNA reagent | pGBKT7 human Titin IgZ4-Z5 | This paper | N/A | Cloning primers in *Supplementary file 1* |
| Recombinant DNA reagent | pGADT7 human MYPN full-length | This paper | N/A | Cloning primers in *Supplementary file 1* |
| Recombinant DNA reagent | pGADT7 human MYPN Ig3-end | This paper | N/A | Cloning primers in *Supplementary file 1* |
| Recombinant DNA reagent | pGADT7 human MYPN Ig5-end | This paper | N/A | Cloning primers in *Supplementary file 1* |
| Recombinant DNA reagent | pGADT7 human MYPN Ig3-4 | This paper | N/A | Cloning primers in *Supplementary file 1* |
| Recombinant DNA reagent | pGADT7 human MYPN Ig4-end | This paper | N/A | Cloning primers in *Supplementary file 1* |
| Recombinant DNA reagent | pGADT7 human PALLD C-term | This paper | N/A | Cloning primers in *Supplementary file 1* |
| Recombinant DNA reagent | pLexA-titin Z3-Z5 | This paper | N/A | Cloning primers in *Supplementary file 1* |
| Recombinant DNA reagent | pLexA-titin Z4-Z5 | This paper | N/A | Cloning primers in *Supplementary file 1* |
| Sequence-based reagent | qRT-PCR primers | This paper | N/A | *Supplementary file 1* |
| Commercial assay or kit | In-Fusion HD Cloning kit | Takara Bio | Cat# 639650 | |
| Commercial assay or kit | Monolith NT Protein Labeling Kit RED-NHS | NanoTemper Technologies | Cat# L001 | |
| Commercial assay or kit | DC Protein Assay Kit II | Bio-Rad Laboratories | Cat# 5000112 | |
| Commercial assay or kit | Frozen-EZ Yeast Transformation II kit | Zymo Research | Cat# T2001 | |

*Appendix 1 Continued on next page*

*Appendix 1 Continued*

| Reagent type (species) or resource | Designation | Source or reference | Identifiers | Additional information |
|---|---|---|---|---|
| Chemical compound, drug | Aureobasidin A | Takara Bio | Cat# 630499 | |
| Chemical compound, drug | X-α-Gal | Takara Bio | Cat# 630,463 | |
| Chemical compound, drug | HisTrap column | GE Healthcare | | |
| Chemical compound, drug | HiLoad 16/600 Superdex 75 pg column | GE Healthcare | Cat# 28989333 | |
| Chemical compound, drug | TRIzol Reagent | Thermo Fisher Scientific | Cat# 15596026 | |
| Chemical compound, drug | High Capacity cDNA Reverse Transcription kit | Thermo Fisher Scientific | Cat# 4368814 | |
| Chemical compound, drug | GoTaq qPCR Master Mix | Promega | Cat# A6002 | |
| Chemical compound, drug | COmplete Protease Inhibitor Cocktail Tablets | Merck Life Science | Cat# 11697498001 | |
| Chemical compound, drug | Pierce Phosphatase Inhibitor Mini Tablets | Thermo Fisher Scientific | Cat# A32957 | |
| Chemical compound, drug | Immobilon Western Chemiluminescent HRP Substrate | Merck Life Science | Cat# WBKLS0500 | |
| Chemical compound, drug | Rhodamine phalloidin | Thermo Fisher Scientific | Cat# R415 | IF (1:100) |
| Chemical compound, drug | Wheat Germ Agglutinin, Alexa Fluor 594 Conjugate | Thermo Fisher Scientific | Cat# W11262 | IF (1:500) |
| Chemical compound, drug | VECTASHIELD Vibrance Antifade Mounting Medium with DAPI | D.B.A. Italia Srl. | Cat# H-1800–10 | |
| Chemical compound, drug | Collagenase, Type 2 | Worthington Biochemical Corporation | Cat# LS004177 | |
| Chemical compound, drug | Fura-2, AM, cell permeant | Thermo Fisher Scientific | Cat# F1201 | |
| Chemical compound, drug | Fluo-4, AM, cell permeant | Thermo Fisher Scientific | Cat# F14201 | |
| Other | 4–20% Criterion TGX Stain Free protein gel,18 well | Bio-Rad Laboratories | Cat# 5678094 | |
| Software, algorithm | Prism, version 7.0 | GraphPad Software Inc | https://www.graphpad.com/scientific-software/prism/ RRID:SCR_002798 | |
| Software, algorithm | Fiji (ImageJ) (analysis software, version 2.0.0-rc-69/1.52 p) | National Institute of Health (NIH) | https://fiji.sc/ RRID:SCR_002285 | SparkMaster plugin used for $Ca^{2+}$ spark analysis |
| Software, algorithm | NT Affinity Analysis software, version 2.0.1334 | NanoTemper Technologies | N/A | |
| Software, algorithm | Ion Wizard, software, version 6.6.11 | IonOptix B.V. | N/A | |

