## [Decision Letter]

**Acceptance summary:**

This study offers important new insight into the role of the cardiac Z-disc protein myopalladin in the response to the pathological stress of pressure overload. While much remains to be learned about myopalladin and other Z-disc structures, emerging data, including the current study, sheds light on their importance in the development of myocardial disease, and their potential for therapeutic targeting. The current work will advance our understanding of cardiac muscle, its normal function, and how the key Z-disc protein myopalladin is involved in heart failure.

**Decision letter after peer review:**

Thank you for submitting your article "Myopalladin knockout mice develop cardiac dilation and show a maladaptive response to mechanical pressure overload" for consideration by *eLife*. Your article has been reviewed by 3 peer reviewers, and the evaluation has been overseen by a Reviewing Editor and Didier Stainier as the Senior Editor. The following individual involved in review of your submission has agreed to reveal their identity: John Konhilas (Reviewer #1).

The reviewers have discussed the reviews with one another and the Reviewing Editor has drafted this decision to help you prepare a revised submission.

As the editors have judged that your manuscript is of interest, but as described below that additional experiments are required before it is published, we would like to draw your attention to changes in our revision policy that we have made in response to COVID-19 (https://elifesciences.org/articles/57162). First, because many researchers have temporarily lost access to the labs, we will give authors as much time as they need to submit revised manuscripts. We are also offering, if you choose, to post the manuscript to bioRxiv (if it is not already there) along with this decision letter and a formal designation that the manuscript is "in revision at eLife". Please let us know if you would like to pursue this option. (If your work is more suitable for medRxiv, you will need to post the preprint yourself, as the mechanisms for us to do so are still in development.)

Summary:

Emerging evidence suggests an essential role of myopalladin, which is a striated muscle-specific sarcomeric protein belonging to a small family of actin-associated Ig domain-containing proteins in the Z-line. Autosomal dominant mutations in MYPN are associated with various types of human cardiomyopathies suggesting a role in cardiac function yet the underlying mechanisms of how MYPN triggers maladaptation in the heart are unknown. The manuscript by Filomena and co-authors describes a novel interaction between titin and myopalladin in the Z-disc, and an interesting maladaptive response to cardiac pressure overload in myopalladin knockout (MKO) mice. The cardiac phenotypes of the MKO mice under basal and stress (pressure overload) conditions suggest that humans with loss-of-function myopalladin mutations may be prone to stress-induced cardiac disease.

Essential revisions:

1. This manuscript would be strengthened by specific and mechanistic insight as to how MYPN triggers aberrant calcium cycling that appears to be secondary to worsening pathology associated with pressure overload.

2. Immunoblots (Figure 6) use both tubulin and GAPDH as loading controls and yet both of these differ significantly following TAC. Clearly, the loading controls are changing but it's not clear if this is a gel loading problem or an actual change in the levels of those proteins. Please reanalyse the Westerns in light of the loading controls. This is likely to significantly alter many of your current findings but there needs to be an adjustment made for protein loading.

3. The authors consistently suggest that CASQ2 is "downregulated" which is incorrect. The data show that in KO mice with TAC at 4W, CASQ2 is not statistically altered relative to the WT control. In fact, there is an upward trend. It may not be increased as much as the WT TAC, but it should not be considered downregulated. The response of CASQ2 to TAC is blunted but not downregulated. Please adjust the paper accordingly, which may change when data is reanalysed using the loading controls.

4. Pg. 6, lines 117-118: The micrograph in Figure 2C does not include good representations of (or any) intercalated discs, so it is impossible to evaluate the authors' statement that intercalated discs are not altered in the MKO.

5. In a cardiomyocyte, calcium levels probably never rise to a level that produces maximum force output and peak calcium concentration is only present for a fraction of the contractile cycle. Thus, the majority of the contraction takes place at submaximal calcium concentrations and inclusion of tension data for the MKO at submaximal calcium concentrations would be more informative. Is calcium sensitivity or cooperativity altered in the MKO mice?

6. The sample sizes (3-4) for the immunoblots to determine titin phosphorylation (Figure 3D) and LV of MKO mice following TAC (Figure 6B) are too low to be confident differences will be detected given the variability evident in some of the blots. Please increase N values or justify the small sample size.

7. Please re-evaluate the statistical analysis of the western blots in Figure 6B. Analysis of the source data provided for pAkt-Thr308/Akt-TAC 4D and CASQ2-TAC 4W indicates that WT and MKO are not significantly different by 2-way ANOVA with a Bonferroni post-hoc test when all groups were considered or Student's t-test when just WT and MKO were compared.

8. Please elaborate in the Discussion on the significance of altered Akt phosphorylation in the MKO mice following TAC.

[Editors' note: further revisions were suggested prior to acceptance, as described below.]

Thank you for submitting your article "Myopalladin knockout mice develop cardiac dilation and show a maladaptive response to mechanical pressure overload" for consideration by *eLife*. Your article has been overseen by a Reviewing Editor and Didier Stainier as the Senior Editor.

Essential revisions:

1) The evaluation of intercalated discs claims no differences between groups and a representative image is provided to support this claim. However, the visual presentation of these single samples from each group appears to show some differences (a broad and diffuse intercalated disc). The authors should measure width or relative area in both WT and KO mice before and after TAC to substantiate the claims in the manuscript.

2) A 2-way ANOVA takes into consideration the interactions of interventions/treatments. Therefore, it is not justifiable to use a Student-T in an isolated case. The statistical analysis should be revised.

---

## [Author Response]

Essential revisions:1. This manuscript would be strengthened by specific and mechanistic insight as to how MYPN triggers aberrant calcium cycling that appears to be secondary to worsening pathology associated with pressure overload.

To provide more mechanistic insights, we performed additional Western blots for intercalated disc proteins and found that SORBS2 is upregulated in MKO mice both under basal conditions and after TAC. Adeno-associated virus 9-mediated overexpression of SORBS2 in mouse heart has been shown cause microtubule densification, T-tubule disorganization, and defective Ca^2+^ handling, (Li et al., EBioMedicine 53, 102695, 2020), including reduced Ca^2+^ amplitude and prolonged time to peak and time to decay of the Ca^2+^ transient in isolated cardiomyocytes. Thus, the altered Ca^2+^ handling in MKO mice may be at least partly explained by the upregulation of SORBS2. SORBS2 is a scaffolding protein located in the intercalated disc and the Z-line, where it interacts with a variety of cytoskeletal proteins, including F-actin, α-actinin and PALLD (Ronty et al., Exp Cell Res. 310, 88-98, 2005; Sanger et al., Cytoskeleton 67, 808-823, 2010). Thus, the upregulation of SORBS2 in MKO mice may be related to alterations in the actin cytoskeleton due to the absence of MYPN. The downregulation of CASQ2 is likely to be a secondary effect of the alterations due to MYPN ablation and SORBS2 upregulation and can explain the reduced Ca^2+^ spark amplitude and width as well as increased Ca^2+^ spark time to peak observed in MKO mice following 7 days of TAC. These considerations have been included in the discussion.

2. Immunoblots (Figure 6) use both tubulin and GAPDH as loading controls and yet both of these differ significantly following TAC. Clearly, the loading controls are changing but it's not clear if this is a gel loading problem or an actual change in the levels of those proteins. Please reanalyse the Westerns in light of the loading controls. This is likely to significantly alter many of your current findings but there needs to be an adjustment made for protein loading.

As described in the “SDS-PAGE and Western blot analysis” section of the “Materials and methods” section in the manuscript, we did not normalize to GAPDH or tubulin since they were not constant after TAC as the reviewer correctly pointed out. Instead the normalization was performed based on total protein content as determined on UV-activated TGX Stain-Free gels (Bio-Rad Laboratories). As described (Gilda and Gomes, Analytical Biochemistry 440, 186-188, 2013; Taylor et al., Mol. Biotechnol. 55, 217-226, 2013; Rivero-Gutiérrez et al., Analytical Biochemistry 467, 1-3, 2014), normalization to total protein content by stain-free staining is a superior method compared to normalization to housekeeping genes, which are not always constant as is the case for tubulin and GAPDH after TAC. The UV-activated TGX Stain-Free gels that were used for the Western blots for GAPDH and tubulin is shown in Author response image 1. As it is evident, the GAPDH and tubulin stainings do not correctly reflect total loaded protein. In particular for the second gel, Western blot analysis using an antibody against pCAMKII, which was not modulated, better reflected total loaded protein than tubulin. To avoid confusion, we have removed GADPH and tubulin from Figure 6K and instead included actin, which does not change after TAC. Also, vinculin, which is a commonly used loading control, is constant before and after TAC.

**Author response image 1. sa2fig1:** 

3. The authors consistently suggest that CASQ2 is "downregulated" which is incorrect. The data show that in KO mice with TAC at 4W, CASQ2 is not statistically altered relative to the WT control. In fact, there is an upward trend. It may not be increased as much as the WT TAC, but it should not be considered downregulated. The response of CASQ2 to TAC is blunted but not downregulated. Please adjust the paper accordingly, which may change when data is reanalysed using the loading controls.

We apologize for the confusion. We repeated the CASQ blots with more replicates and found that CASQ2 is increased in both WT and MKO mice 4 weeks after TAC, while CASQ2 is significantly reduced in MKO mice compared to WT mice 4 days after TAC. Due to a 3- to 4-fold upregulation of CASQ2 4 weeks after TAC, this difference is not significant by 2-way ANOVA, but highly significant by Student’s t-test (p = 0.0017). Since CASQ2 is not significantly upregulated in WT mice 4 days after TAC, we have continued to use “downregulated” throughout the manuscript when referring to the 4-day time point.

The text has been modified accordingly.

“Western blot analyses for proteins involved in calcium handling revealed increased CASQ2 levels 4 weeks after TAC both in WT and MKO mice, but significantly reduced CASQ2 levels in MKO mice compared to WT mice 4 days after TAC (∼2.5-fold, p = 0.0017; Student’s t-test) (Figure 6K, N). Due to a 3- to 4-fold upregulation of CASQ2 4 weeks after TAC, the difference in CASQ2 expression between MKO and WT mice 4 days after TAC was not statistically significant by 2-way ANOVA (Figure 6N). At the transcript level, Casq2 was downregulated in WT mice 4 days after TAC, but not 4 weeks after TAC, while it was unchanged in MKO mice (Figure 6J). This suggests that CASQ2 is regulated at the post-transcriptional level.”

4. Pg. 6, lines 117-118: The micrograph in Figure 2C does not include good representations of (or any) intercalated discs, so it is impossible to evaluate the authors' statement that intercalated discs are not altered in the MKO.

We have now included representative pictures of the intercalated disc in *Figure 2H* and *5F*. We found no obvious changes in the intercalated disc in MKO mice either under basal conditions or 4 weeks after TAC. To search for possible changes in expression levels of intercalated disc proteins, we performed Western blot analyses and found upregulation of SORBS2/ArgBP2 in MKO mice both under basal conditions and after TAC as well as upregulation of desmoplakin in MKO mice 4 days after TAC (*Figure 6K, O, P*). This has been described in the following section:

“On the other hand, Western blot analysis for components of the intercalated disc, including desmosomal (desmoplakin, plakoglobin/γ-catenin), adherens junction (N-cadherin, α-E-catenin, β-catenin, plakoglobin/γ-catenin, vinculin, sorbin and SH3 domain-containing 2 (SORBS2, also known as Arg Binding Protein 2 (ArgBP2)), and gap junction (connexin 43) proteins, showed upregulation of desmoplakin 4 days after TAC (∼2.3-fold; Figure 6K, D) and SORBS2), a known interaction partner of PALLD (Ronty et al., 2005), both 4 days (1.8-fold) and 4 weeks (∼1.6-fold) after TAC in MKO mice compared to WT mice (Figure 6K, E). SORBS2 was also significantly upregulated in MKO mice under basal conditions by Student’s t-test (∼1.6-fold, p = 0.0078), but as SORBS2 was nearly 4-fold induced after TAC in MKO mice the difference was not statistically significant by 2-way ANOVA. The SORBS2 interaction site in PALLD is located in a proline-rich region, not present in MYPN (Ronty et al., 2005). Correspondently, MYPN did not bind to SORBS2 in the yeast two-hybrid system (data not shown).”

5. In a cardiomyocyte, calcium levels probably never rise to a level that produces maximum force output and peak calcium concentration is only present for a fraction of the contractile cycle. Thus, the majority of the contraction takes place at submaximal calcium concentrations and inclusion of tension data for the MKO at submaximal calcium concentrations would be more informative. Is calcium sensitivity or cooperativity altered in the MKO mice?

Results from calcium jump experiments at submaximal activation have now been included in Figure 3M. These experiments indicate that calcium sensitivity is not altered in the MKO mice and suggest that the defect in force generation at maximal calcium activation is also present at submaximal calcium concentrations. This has been added to the manuscript as detailed below.

“Furthermore, measurements of the tension generated at submaximal (pCa 5.75) vs. maximal (pCa 4.50) Ca^2+^-activation in Ca^2+^-jump experiments showed a similar ratio in WT and MKO myofibrils (Figure 3M), providing strong evidence that the lack of MYPN does not affect myofilament Ca^2+^-sensitivity.”

6. The sample sizes (3-4) for the immunoblots to determine titin phosphorylation (Figure 3D) and LV of MKO mice following TAC (Figure 6B) are too low to be confident differences will be detected given the variability evident in some of the blots. Please increase N values or justify the small sample size.

As requested, we repeated the Western blots in Figures 3O-S, Figure 3—figure supplement 2, and Figure 6K-P with more replicates. The upregulation of CASQ2 4 weeks after TAC was confirmed, but there was no significant difference between WT and MKO mice. In contrast, CASQ2 was significantly reduced in MKO mice compared to WT mice 4 days after TAC (p = 0.0017; Student’s t-test; see Figure 6K, N and the Author response image 2). Due to a 3- to 4-fold upregulation of CASQ2 4 weeks after TAC, the difference was not statistically significant by 2-way ANOVA with Bonferroni’s post-hoc test.

The reduced expression of pAkt-Ser473/Akt in MKO mice after TAC was not confirmed (see Author response image 2) and when we repeated the Western blots with the pAkt-308 antibody we realized that the band did not have the correct size compared to total Akt antibody. We then tested several different pAkt-308 antibodies, but were unable to get a signal at the correct size in the heart with any of the antibodies. From the vendors, we were told that the pAkt-308 is more difficult to detect than pAkt-473, so it is likely that the expression is simply not high enough for detection in the heart. For this reason, we decided to exclude the Western blot for pAkt-308. Since pAkt was not altered we also didn’t include the Western blots for the downstream targets GSK3β and 4E-BP1, also because their phosphorylation levels were highly variability when we repeated the Western blots with additional samples, making it difficult to obtain conclusive results. Instead, we performed additional Western blots for components of the desmosome, adherens junction, and gap junction of the intercalated disc, which revealed upregulation of SORBS2 in MKO mice both under basal conditions and after TAC as well as upregulation of desmoplakin in MKO mice 4 days after TAC (Figure 6K, O, P). We also included a Western blot for MYPN’s interaction partner MRTF-A, but it wasn’t modulated. Several of the blots in Figure 6K and Figure 3—figure supplement 2 have been replaced with more representative blots.

7. Please re-evaluate the statistical analysis of the western blots in Figure 6B. Analysis of the source data provided for pAkt-Thr308/Akt-TAC 4D and CASQ2-TAC 4W indicates that WT and MKO are not significantly different by 2-way ANOVA with a Bonferroni post-hoc test when all groups were considered or Student's t-test when just WT and MKO were compared.

We have redone all the statistical analyses with additional replicates using 2-way ANOVA with Bonferroni’s post-hoc test. As explained above, CASQ2 is significantly reduced in MKO mice compared to WT mice 4 days after TAC by Student’s t-test, but not by 2-way ANOVA with Bonferroni’s post-hoc test due to a 3- to 4-fold upregulation of CASQ2 4 weeks after TAC. Similarly, SORBS2 is upregulated in MKO mice under basal conditions by Student’s t-test, but not by 2-way ANOVA with Bonferroni’s post-hoc test as SORBS2 is highly upregulated in MKO mice after TAC.

8. Please elaborate in the Discussion on the significance of altered Akt phosphorylation in the MKO mice following TAC.

As explained above, this is not anymore relevant as we didn’t confirm the altered Akt phosphorylation when repeating the Western blots with more replicates.

[Editors' note: further revisions were suggested prior to acceptance, as described below.]

Essential revisions:1) The evaluation of intercalated discs claims no differences between groups and a representative image is provided to support this claim. However, the visual presentation of these single samples from each group appears to show some differences (a broad and diffuse intercalated disc). The authors should measure width or relative area in both WT and KO mice before and after TAC to substantiate the claims in the manuscript.

We thank the reviewers for this perceptive observation and valuable suggestion. As suggested, we have now measured intercalated disc (ICD) fold amplitude under basal conditions and following TAC. The details are described in the revised Materials and methods section. To our surprise, this revealed a ∼27% increase in ICD fold amplitude in MKO mice under basal conditions (see Figure 2I). Furthermore, after TAC MKO mice showed a ∼70% increase in ICD fold amplitude compared to WT mice, while no significant increase in ICD fold amplitude was observed in WT mice after TAC (see Figure 5G). According to this finding, we have modified the discussion, where we mention that increased ICD fold amplitude is frequently associated with dilated cardiomyopathy. Furthermore, we have modified the abstract and the schematic illustration of the phenotype of MKO mice in Figure 8. We very much appreciate this request by the reviewers as we believe that this finding has significantly broadened the manuscript.

2) A 2-way ANOVA takes into consideration the interactions of interventions/treatments. Therefore, it is not justifiable to use a Student-T in an isolated case. The statistical analysis should be revised.

To solve this issue, we involved a bioinformatician, Dr. Simone Serio, who performed a statistical review of the complete manuscript and has been included among the authors. We first determined whether ANOVA assumptions were met using the Shapiro-Wilk test for normality within each group in the various analyses. In addition, we tested if residuals were normally distributed, because in some cases sample groups in the same experiment were significantly different from normality, while others were not. The equality of variance across groups was verified using Bartlett’s test, while the heteroscedasticity of residuals was determined using Spearman's rank correlation test. In cases where one or more of these tests failed, data were log-transformed to meet ANOVA assumptions. For the statistical analysis of echocardiographic parameters in WT and MKO mice (the between-subjects factor) before and after TAC (the within-subjects factor) in Figure 4A-H, a linear mixed model (LMM) was used.

We previously analyzed the SORBS2 and CASQ2 densitometry data by two-way ANOVA, but our statistical analysis showed that the ANOVA assumptions had been violated. Log-transformation of the data solved this issue after which two-way ANOVA was performed. This showed significantly increased SORBS2 expression both under basal conditions and after short- and long-term TAC, while CASQ2 is significantly decreased 4 days after TAC. The isolated Student’s t-tests in the text have been removed and the figures have been modified with the corrected statistical analyses.